# DUAL-PRIV PRUNING: EFFICIENT DIFFERENTIAL PRIVATE FINE-TUNING IN MULTIMODAL LARGE LANGUAGE MODELS

## ABSTRACT

Differential Privacy (DP) is a widely adopted technique, valued for its effectiveness in protecting the privacy of task-specific datasets, making it a critical tool for large language models. However, its effectiveness in Multimodal Large Language Models (MLLMs) remains uncertain. Applying Differential Privacy (DP) inherently introduces substantial computation overhead, a concern particularly relevant for MLLMs which process extensive textual and visual data. Furthermore, a critical challenge of DP is that the injected noise, necessary for privacy, scales with parameter dimensionality, leading to pronounced model degradation; This trade-off between privacy and utility complicates the application of Differential Privacy (DP) to complex architectures like MLLMs. To address these, we propose **Dual-Priv Pruning**, a framework that employs two complementary pruning mechanisms for DP fine-tuning in MLLMs: (i) *visual token pruning* to reduce input dimensionality by removing redundant visual information, and (ii) *gradient-update pruning* during the DP optimization process. This second mechanism selectively prunes parameter updates based on the magnitude of noisy gradients, aiming to mitigate noise impact and improve utility. Experiments demonstrate that our approach achieves competitive results with minimal performance degradation. In terms of computational efficiency, our approach consistently utilizes less memory than standard DP-SGD. While requiring only 1.74% more memory than zeroth-order methods which suffer from severe performance issues on A100 GPUs, our method demonstrates leading memory efficiency on H20 GPUs. To the best of our knowledge, we are the first to explore DP fine-tuning in MLLMs. Our code is avaliable in : `https://anonymous.4open.science/r/Dual-priv-pruning-AE7E`.

## 1 INTRODUCTION

Large Language Models (LLMs) Zhang et al. (2022); Radford et al. (2019); Touvron et al. (2023) have showcased remarkable proficiency in natural language processing, driving their widespread adoption in downstream tasks Ziegler et al. (2019), and Multimodal Large Language Models (MLLMs) Liu et al. (2023); Wang et al. (2024a); Abdin et al. (2024)extend the power of LLMs by integrating text and visual data, opening up possibilities for applications that require understanding across different modalities. However, both models are easy to risk leaking sensitive information during training Das et al. (2025); Meskó (2023). Differential Privacy Dwork (2006) (DP) , the technology for providing privacy guarantees that limit the ability to infer whether a data point was used in the training process of a model by observing its output. This technology is typically achieved by injecting noise during training processes, limiting the discernible impact of single data point. The degree of privacy guarantee is tuned using a privacy budget ($\epsilon$), where stronger privacy guarantee (lower $\epsilon$) generally comes at the cost of adding more noise and degrading model performance. The inherent trade-off between privacy and utility presents a significant challenge, particularly when applying DP to large and complex models like LLMs, since the necessary noise often scales with parameter dimensionality. Prior works Yu et al. (2021a); Li et al. (2021); Liu et al. (2024); Goel et al. (2025) have shown that LLMs with hundreds of millions of parameters can be effectively and efficiently fine-tuned to yield models with high performance under modest privacy leakage.

However, it remains unclear whether such conclusions of LLMs are transferable to MLLMs. Similar to unimodal models, DP also face challenges under MLLMs. The first is **computation consumption**. This challenge is exacerbated in MLLMs, which rely on a large number of visual tokens ( e.g., 197 tokens per image in CLIP-ViT Radford et al. (2021) or hundreds in LLaVA Liu et al. (2023) ) to represent detailed visual information, significantly increasing computation demands. Recent work Li et al. (2021) introduced "ghost clipping" as an efficient technique to reduce the computational cost of DP-SGD, which is in principle applicable to both sequential language models and non-sequential architectures. Nevertheless, in multimodal large language models the coexistence of heterogeneous modules (e.g., ViTs, projectors) imposes substantial practical barriers for ghost clipping and thus a significant limitation in real-world MLLM settings. Zeroth-order methods (e.g., DP-ZO Tang et al. (2024)) also aim to reduce computation overhead by avoiding explicit gradient calculations. However, these methods introduce severe convergence issues. For instance, DP-ZO required more training steps (75k vs 200) than standard DP-SGD to achieve comparable performance on SQuAD Tang et al. (2024), making this gradient-free approach prohibitively slow for practical MLLM training. Another challenge is **model degradation**. Differential privacy introduces noise to safeguard data privacy, but this noise perturbs the gradient signals during training, leading to performance degradation. In MLLMs, DP noise scales with parameter dimensionality, overwhelming gradient signals in high-dimensional layers and necessitating more iterations to stabilize optimization, as noted in foundational work on DP-SGD Abadi et al. (2016).

To tackle these challenges, we introduce **Dual-Priv Pruning**, a novel DP finetuning approach tailored for MLLMs. Our approach integrates two complementary pruning mechanisms designed to work in concert, addressing these issues from both the input representation and the optimization process. The first key pruning mechanism focuses on optimizing the visual input stream prior to training: it employs an attention-based mechanism to identify and prune redundant visual tokens, thereby substantially reducing the input dimensionality and subsequent computational demands. The less critical visual information pruned in this manner is then fused into some compact contextual representations, to which a calibrated heuristic noise is added. This step aims to preserve essential global context while further alleviating the processing load for the differential privacy mechanism. The second core pruning mechanism refines the differential private fine-tuning process itself. While adhering to the standard DP-SGD framework for rigorous noise addition to guarantee privacy, Dual-Priv Pruning introduces a *gradient-update pruning* technique. This technique analyzes the noisy gradients resulting from DP noise injection. It then selectively applies these gradients for parameter updates only to those blocks where the underlying signal is deemed sufficiently strong and reliable to overcome the obfuscating effect of the DP noise, thereby preserving model utility and stabilizing training. Dual-Priv Pruning offers a robust solution. As the first work to explore DP finetuning specifically tailored for MLLMs, our method achieves a superior privacy-utility trade-off and enhanced computational efficiency, delivering competitive performance even under stringent privacy budgets.

We summarize our main contributions as follows: **(1)** We pioneer the integration of DP into the domain of MLLMs, addressing a critical research gap in privacy-preserving multimodal learning. **(2)** We introduce a novel privacy-aware visual pruning mechanism that significantly reduces computational overhead by optimizing visual inputs, thereby creating more favorable conditions for subsequent DP fine-tuning. **(3)** We propose an DP-compatible gradient-update pruning strategy that intelligently applies noisy gradients to mitigate the adverse effects of DP noise on model performance, thereby enhancing utility while maintaining strong privacy guarantees. **(4)** Extensive experiments demonstrate that our Dual-Priv Pruning achieves robust privacy protection, substantial memory reduction, and competitive performance on diverse vision-language tasks, even under stringent privacy budgets.

## 2  RELATED WORK

**Differential Privacy (DP)** Dwork (2006) ensures privacy guarantees by limiting the ability to infer whether a data point was used in the training process of a model, making it a cornerstone for privacy-preserving learning. In the area of computer vision, Tang et al. (2023a) developed DP methods for image classification by adding noisy priors, achieving strong privacy-utility trade-offs, and Luo et al. (2024) applied DP to video recognition, enforcing video-level differential privacy through clip-based classification models. In natural language processing, McMahan et al. (2017) trained recurrent language models with DP, reducing risks of data memorization. For LLMs, Li et al. (2024a) demonstrated DP fine-tuning but noted challenges with utility degradation due to noise, while Kerrigan et al. (2020) showed that public pre-training followed by private fine-tuning can alleviate

some performance losses. Memory-efficient techniques, such as "ghost-clipping" Li et al. (2021), optimize DP-SGD for LLMs but rely on text-specific assumptions, limiting their applicability to multimodal settings. Zeroth-order optimization Tang et al. (2024) offers an alternative for LLMs by avoiding gradient instantiation, but it suffers from too long training times. Other efforts to improve DP include manipulating gradients, such as GIP Yang et al. that perturbed individual gradient indices, though its privacy analysis clarity was questioned; In contrast, our gradient-update pruning operates as a post-processing step on entire noised logical parameter blocks, simplifying privacy analysis and aligning with PEFT. In multimodal learning, Huang et al. (2023) introduced DP to CLIP training, protecting vision-language data, and Yu et al. (2021b) proposed low-rank reparametrization for scalable private learning, applicable to multimodal tasks. Additionally, Kaissis et al. (2021) applied DP to medical image, emphasizing privacy in sensitive domains. Despite these advances, no prior work has explored DP fine-tuning for MLLMs, which face unique challenges due to cross-modal interactions and massive length visual tokens. Existing methods, do not address the memory demands and model degradation of MLLMs, a gap that our work to addresses.

**Multimodal Large Language Models (MLLMs)** integrate visual and textual modalities to solve a wide range of tasks. Flamingo Alayrac et al. (2022) introduced a query-based cross-attention mechanism to enable multimodal interactions, while BLIP-2 Li et al. (2023b) proposed the lightweight Q-Former to enhance efficiency. InstructBLIP Dai et al. (2023) further aligned models with user intent via instruction tuning across diverse datasets. LLaVA Liu et al. (2023) improved visual understanding using curated training data, while subsequent efforts such as Qwen-VL Bai et al. (2023) and CogVLM Wang et al. (2024b) introduced advanced training strategies and modular visual expert systems to boost performance. A major challenge in MLLMs is the redundancy of visual tokens, which significantly increases memory and computational costs Chen et al. (2024). Recent work addresses this inefficiency: FastVLM Vasu et al. (2024) prunes tokens based on attention scores, and VisionZip Yang et al. (2024) identifies contextual tokens that retain global semantics (e.g., background information). Visual token redundancy offers a promising avenue for DP in MLLMs. Pruning low-importance tokens reduces sensitive data exposure. We leverage this property to enable even source-level privacy protection and efficient DP fine-tuning.

## 3 PRELIMINARY

### 3.1 DIFFERENTIAL PRIVACY

Differential privacy (DP) Dwork (2006) provides a rigorous framework to safeguard sensitive data by ensuring that model outputs remain statistically indistinguishable for datasets differing by a single record. This guarantee inherently limits the ability of inferring individual record participation, mitigating risks such as membership inference attacks Shokri et al. (2017). A hallmark of DP is its robustness to post-processing: if an algorithm $\mathcal{A}$ satisfies $(\epsilon, \delta)$-DP, any function $f \circ \mathcal{A}$ preserves the same $(\epsilon, \delta)$-DP guarantee.

**Definition 1** (($\epsilon, \delta$)-Differential Privacy). *A randomized algorithm $\mathcal{A}$ is $(\epsilon, \delta)$-differentially private if, for any two neighboring datasets $\mathcal{D}$ and $\mathcal{D}'$, differing by one record, and any set of outputs $S \subseteq Range(\mathcal{A})$, the following holds:*

$$\Pr[\mathcal{A}(\mathcal{D}) \in S] \leq e^\epsilon \cdot \Pr[\mathcal{A}(\mathcal{D}') \in S] + \delta, \tag{1}$$

*where $\epsilon \geq 0$ is the privacy budget, controlling the strength of the privacy guarantee, and $\delta \in [0, 1)$ is a small failure probability.*

In the context of fine-tuning MLLMs, two datasets $\mathcal{D}$ and $\mathcal{D}'$ are defined as neighboring if one can be obtained from the other by adding or removing a single image-text pair. The application of DP in iterative training (introduced in Section 3.1.1), relies on fundamental mechanisms and accounting principles. The Gaussian Mechanism (detailed in Fact A.1) is employed to add noise. To manage the overall privacy loss across multiple iterations, privacy accounting techniques like Rényi Differential Privacy (RDP) (detailed in Fact A.2) are utilized. These principles are central to the DP application.

### 3.1.1 DIFFERENTIALLY PRIVATE SGD

Differentially Private Stochastic Gradient Descent (DP-SGD) Abadi et al. (2016) adapts SGD to ensure the trained model parameters $\theta \in \mathbb{R}^d$ satisfy an overall $(\epsilon, \delta)$-DP guarantee with respect to $\mathcal{D}_{\text{train}}$. In each iteration $k$, for a minibatch $\xi_k$ of size $m$ sampled with probability $q = m/N$: First, per-sample gradients $g_i = \nabla_\theta \mathcal{L}(\theta_{k-1}, (\mathcal{I}_i, \mathcal{T}_i))$ are computed for each $i \in \xi_k$. Second, to bound sensitivity, the $L_2$ norm of each gradient $g_i$ is clipped using a threshold $C$: $\hat{g}_i = g_i / \max(1, \|g_i\|_2 / C)$.

This ensures $\|\hat{g}_i\|_2 \leq C$, thereby limiting the influence of any single sample and resulting in an $L_2$ sensitivity of $\Delta f = C/m$ for the subsequent average gradient (details in Appendix B). Third, these clipped gradients are aggregated by averaging: $\bar{g} = \frac{1}{m}\sum_{i \in \xi_k} \hat{g}_i$. Finally, calibrated Gaussian noise is added to this average gradient before updating:

$$\theta_k = \theta_{k-1} - \eta \cdot \left( \bar{g} + \mathcal{N}(0, \sigma^2 C^2 I_d/m^2) \right). \tag{2}$$

The hyperparameters $C$ (clipping norm) and $\sigma$ (noise multiplier) control the trade-off between privacy and utility. The appropriate value for $\sigma$ is determined based on the overall privacy budget $(\epsilon, \delta)$, total training steps, and sampling rate, typically using privacy accounting methods like RDP (Fact A.2).

## 3.2 PROBLEM DEFINITION: DIFFERENTIALLY PRIVATE FINE-TUNING OF MLLMs

Our work focuses on fine-tuning a pre-trained MLLM $\mathcal{M}_\theta$ with parameters $\theta \in \mathbb{R}^d$. The fine-tuning is performed on a private dataset $\mathcal{D}_{\text{fine}} = \{(\mathcal{I}_i, \mathcal{T}_i)\}_{i=1}^N$, where each pair consists of an image $\mathcal{I}_i$ and a text sequence $\mathcal{T}_i = \{w_1, \ldots, w_i\}$. The primary objective is to adapt $\mathcal{M}_\theta$ to downstream vision-language tasks by learning parameters $\theta_{\text{fine}}$ that exhibit **high utility**. This utility is typically achieved by minimizing an empirical risk, often the negative log-likelihood loss, over the $\mathcal{D}_{\text{fine}}$.

A crucial and defining requirement for this process is that it must adhere to a **strict $(\epsilon, \delta)$-Differential Privacy (DP) guarantee** (Definition 1) with respect to $\mathcal{D}_{\text{fine}}$. This requires the learning algorithm $\mathcal{A}$ to generate $\theta_{\text{fine}}$ from $\mathcal{D}_{\text{fine}}$ and $\theta$ under $(\epsilon, \delta)$-DP guarantees. The core problem can be summarized as finding parameters $\theta_{\text{fine}}$ that balance utility and privacy, as formally stated below:

---

**Problem Formulation**

**Objective:** Minimize the empirical risk on the private dataset $\mathcal{D}_{\text{fine}}$:

$$\mathcal{L}(\theta, \mathcal{D}_{\text{fine}}) := \frac{1}{N} \sum_{i=1}^N \left( -\sum_{t=1}^{T_i} \log P_{\mathcal{M}_\theta}(w_{i,t} | \mathcal{I}_i, w_{i,1}, \ldots, w_{i,t-1}) \right) \tag{3}$$

The learning algorithm $\mathcal{A}$ producing $\theta_{\text{fine}}$ from $\mathcal{D}_{\text{fine}}$ must be $(\epsilon, \delta)$-Differentially Private:

$$\text{Find } \theta_{\text{fine}} \approx \arg\min_{\theta \in \mathbb{R}^d} \mathcal{L}(\theta, \mathcal{D}_{\text{fine}}) \quad \text{s.t.} \quad \mathcal{A}(\mathcal{D}_{\text{fine}}) \text{ is } (\epsilon, \delta)\text{-DP.} \tag{4}$$

---

## 4 METHOD

We introduce **Dual-Priv Pruning**, the first framework for differential private (DP) fine-tuning of MLLMs, designed to optimize the privacy-utility trade-off. **Mechanism 1** performs attention-based token pruning and fusion to transform visual input into compact representation $\mathcal{V}'$. Motivated by evidence that token *quality* (not quantity) drives VLM utility and that many visual tokens are redundant or misaligned, we use [CLS] attention as a prompt-invariant signal to retain information-dense tokens and fuse non-dominant ones (Yang et al., 2024; Vasu et al., 2024; Shang et al., 2024; Zhang et al., 2024b). The pruning decision depends on visual tokens (text-agnostic), which shortens sequences and reduces compute without consuming the DP budget. See more details in Appendix O. **Mechanism 2** applies $(\epsilon, \delta)$-DP to trainable parameters $\theta_{\text{train}}$ using DP-SGD (Section 3.1.1), enhanced with a *gradient-update pruning* strategy to improve utility. This provides formal $(\epsilon, \delta)$-DP guarantees for the entire pipeline. Further details and motivations are in Appendix E, Appendix F, Appendix M and Appendix O

## 4.1 MECHANISM 1: VISUAL TOKEN PRUNING AND FUSION

This initial stage reduces the computation cost associated with long visual token sequences before the differential private fine-tuning process begins. It consists of identifying and retaining the most important visual tokens based on attention, followed by merging the remaining tokens and applying noise prior. This stage is not designed to provide the formal DP guarantee.

**Dominant Token Selection via CLS Attention.** For an input image $\mathcal{I}_i$, the vision encoder extracts an initial set of $n$ visual tokens $\mathcal{V} = \{v_{cls}, v_1, \ldots, v_{n-1}\}$, including a class token $v_{cls}$ and $n-1$ patch tokens, where $v_j \in \mathbb{R}^d$. We hypothesize that tokens receiving significant attention from the class token ([CLS]) include the most critical global information.

To identify these dominant tokens, we first compute the multi-head self-attention maps within a selected layer of the vision encoder. The attention map for a single head $h$ is given by:

Figure 1: Overview of our Dual-Priv Pruning. **(Left)**: Visual Token Pruning and Fusion. Using [CLS] attention, dominant tokens are selected; less important ones are averaged with heuristic noise. **(Right)**: DP Fine-tuning with gradient pruning. Noise is added to gradients in LLM blocks, and updates are selectively applied based on noisy gradient magnitude. Frozen parameters remain unchanged.

$$S_h = \text{Softmax}\left(\frac{Q_h K_h^\top}{\sqrt{D_h}}\right) \in \mathbb{R}^{n \times n}, \tag{5}$$

where $Q_h, K_h$ are the query and key matrices, and $D_h$ is the head dimension. We average these maps across all $H$ heads to get an aggregated attention map $S_{\text{avg}} \in \mathbb{R}^{n \times n}$:

$$S_{\text{avg}} = \frac{1}{H} \sum_{h=1}^{H} S_h. \tag{6}$$

The importance score $s_j$ for each patch token $v_j$ ($j \in \{1, \ldots, n-1\}$) is then determined by the attention receives from the [CLS] token in the aggregated map. We select the $K$ patch tokens with the highest importance scores $s_j$ as the dominant patch tokens $\mathcal{V}_d = \{v_j \mid s_j \text{ is among the top K scores}\}$. The class token $v_{cls}$ is always retained. The remaining patch tokens form the non-dominant set $\mathcal{V}_{nd}$.

**Contextual Token Fusion and Heuristic Noise.** To preserve the visual context features from $\mathcal{V}_{nd}$ while reducing sequence length, we uniformly randomly select tokens $v_{\text{center},i}$ from $\mathcal{V}_{nd}$ as cluster centers and enhance their representation based on cosine similarity with the remaining non-dominant tokens. Subsequently, Gaussian noise scaled by $\sigma_{\text{fuse}}^2$ is heuristically applied to the enhanced $v_{\text{center}}$, producing the fused contextual tokens $c$, as defined in the following formula:

$$\mathbf{c} = \begin{bmatrix} v_{\text{center},1} + \frac{1}{|\mathcal{C}_1|} \sum_{v_j \in \mathcal{C}_1} v_j \\ v_{\text{center},2} + \frac{1}{|\mathcal{C}_2|} \sum_{v_j \in \mathcal{C}_2} v_j \\ \vdots \\ v_{\text{center},k} + \frac{1}{|\mathcal{C}_k|} \sum_{v_j \in \mathcal{C}_k} v_j \end{bmatrix} + \mathcal{N}\left(0, \sigma_{\text{fuse}}^2 I_{kd}\right), \tag{7}$$

where $\mathcal{C}_i$ is the set of non-dominant tokens assigned to the $i$-th cluster based on similarity:

$$\mathcal{C}_i = \left\{ v_j \in \mathcal{V}_{nd} \mid i = \arg\max_l \text{sim}(v_j, v_{\text{center},l}) \right\}, \quad i = 1, 2, \ldots, k. \tag{8}$$

The noise adding process serves as a form of regularization or stochasticity injection; A key aspect of our design is to maintain consistency with the noise introduced by the DP mechanism in the subsequent stage. Therefore, the variance of this heuristic noise, $\sigma_{\text{fuse}}^2$, is set to be equivalent to the variance of the Gaussian noise added per step in the DP optimization process (Mechanism 2, Section 4.2). It does not contribute to the formal $(\epsilon, \delta)$-DP guarantee derived in Mechanism 2. The final set of visual tokens passed to the MLLM for the DP fine-tuning stage is $\mathcal{V}' = \{v_{cls}\} \cup \mathcal{V}_d \cup \{C\}$, which has a significantly reduced size of $K + |C| + 1$ tokens.

## 4.2 MECHANISM 2: DP FINE-TUNING WITH GRADIENT-UPDATE PRUNING

This core mechanism performs the $(\epsilon, \delta)$-differential private fine-tuning of the trainable parameters $\theta_{\text{train}}$ (e.g., LoRA matrices Hu et al. (2022)), leveraging the pruned visual inputs $(\mathcal{V}', \mathcal{T})$ from Mechanism 2. Our approach builds upon DP-SGD (Section 3.1.1) but introduces a **post-noise adaptive update** mechanism designed to enhance utility without compromising the privacy guarantee.

The process within each training iteration $t$ begins with standard DP-SGD procedures. For a minibatch $\xi_t$ of size $m$, we first compute per-sample gradients $g_i = \nabla_{\theta_{\text{train}}} \mathcal{L}(\theta_{t-1}; (\mathcal{V}'_i, \mathcal{T}_i))$. To bound the influence of individual samples, we clip the $L_2$ norm of each gradient using a threshold $C$: $\hat{g}_i = g_i / \max(1, \|g_i\|_2 / C)$. These clipped gradients are then averaged across the minibatch to produce $\bar{\hat{g}} = \frac{1}{m} \sum_{i \in \xi_t} \hat{g}_i$. The crucial step for ensuring differential privacy follows. Gaussian noise is added *unconditionally* to the entire aggregated gradient vector:

$$\tilde{g} = \bar{\hat{g}} + \mathcal{N}\left(0, \frac{\sigma^2 C^2}{m^2} I_{d_{\text{train}}}\right). \tag{9}$$

Here, $d_{\text{train}}$ is the dimensionality of $\theta_{\text{train}}$, and the noise multiplier $\sigma$ is determined by the overall privacy budget $(\epsilon, \delta)$, number of steps $T$, and sampling rate $q$ via privacy accounting (Fact A.2). At this point, the noisy gradient $\tilde{g}$ is an $(\epsilon_t, \delta_t)$-differentially private quantity for the current step. Our mechanism diverges from standard DP-SGD hereafter. Instead of directly using $\tilde{g}$ for the update, we first analyze its structure and magnitude. We partition $\tilde{g}$ into components $\tilde{g}_j$ corresponding to logical parameter blocks within $\theta_{\text{train}}$ and compute the $L_2$ norm $N_j = \|\tilde{g}_j\|_2$ for each block.

Based on these norms, we generate a binary mask $M$, structured identically to $\theta_{\text{train}}$, to selectively prune the parameter update. A block $j$ is chosen for update ($M_j$ remains 1): **only if its noisy gradient norm $N_j$ is among the top K% of norms across all blocks**, otherwise $M_j$ remains 0.

$$M_j = \mathbb{I}(N_j \in \text{Top-K\%}(\{N_1, N_2, \ldots, N_J\})), \tag{10}$$

where $\mathbb{I}(\cdot)$ is the indicator function, $J$ is the total number of parameter blocks, and Top-K%$(\cdot)$ denotes the set of the $K\%$ largest norm values. The percentage for K% is a hyperparameter.

Finally, the model parameters are updated using the noisy gradient $\tilde{g}$, but applied selectively through the generated mask $M$ via element-wise multiplication (Hadamard product $\odot$):

$$\theta_t = \theta_{t-1} - \eta_t \cdot (M \odot \tilde{g}). \tag{11}$$

This ensures that parameter updates are only applied to blocks where the noisy gradient signal was deemed sufficiently strong or reliable according to the gating criterion. The full step-by-step procedure is formally detailed in Section L.

### 4.3 Overall Privacy Guarantee

The $(\epsilon, \delta)$-DP guarantee of the Dual-Priv Pruning method is entirely derived from Mechanism 2 (Section 4.2). Mechanism 1 (Section 4.1) involves data preprocessing *before* the DP mechanism is applied and does not consume privacy budget. The adaptive update mechanism within Mechanism 2, constitutes post-processing on the private intermediate result $\tilde{g}$ and thus does not affect the formal $(\epsilon, \delta)$-DP guarantee (Section D).

## 5 Experiments

We conduct a comprehensive experimental evaluation of our proposed **Dual-Priv Pruning** method. Our experiments are designed to validate four core advantages of Dual-Priv Pruning: **(1)** Preserve utility, especially under strict privacy budgets ($\epsilon \leq 3$), compared to baseline methods; **(2)** Significant improvements in computation cost, highlighted by an approximate 14.34% reduction in peak GPU memory usage; and **(3)** Validated effectiveness on challenging, visual tasks, encompassing high-resolution real-world scenes and medical images, demonstrating the method's practical applicability in complex, privacy-sensitive domains. **(4)** Empirically shown to be effective against privacy attacks like Membership Inference Attacks (MIA).

### 5.1 Experimental Setup

**Datasets.** We evaluate performance by fine-tuning on the training sets and evaluating on the test sets of several vision-language benchmarks. These include standard datasets such as ScienceQA Lu et al. (2022) (Scientific VQA), TextVQA Singh et al. (2019) (VQA over text in images), and GQA Hudson & Manning (2019) (Compositional VQA). To specifically assess scalability and robustness on complex inputs, we utilize MME-RealWorld Zhang et al. (2024a), an MLLM benchmark designed for high-difficulty tasks involving high-resolution real-world images. Additionally, we incorporate two medical visual question answering dataset, PathVQA He et al. (2020)and VQA-RAD Lau et al. (2018), to further test generalization on specialized, challenging domains.

**Model & Training Strategy.** We utilize LLAVA-7B Liu et al. (2023) as our base MLLM. Specifically, for tasks in the medical domain (PathVQA, VQA-RAD, and MIA on ROCOV2), we employ Med-LLaVALi et al. (2023a), a LLaVA variant adapted for medical vision-language understanding. To

Table 1: Comparison of different methods on standard benchmarks (BS = 12). For reference, non-private performance ($\epsilon = \infty$) are included. Metrics reported are Accuracy (Acc) and Image-based Accuracy (IMG). The best results for each $\epsilon$ setting are shown in **bold**.

| | DZPO | | | | DP-SGD | | | | Dual-Priv(ours) | | | |
|---|---|---|---|---|---|---|---|---|---|---|---|---|
| | ScienceQA | | TextVQA | GQA | ScienceQA | | TextVQA | GQA | ScienceQA | | TextVQA | GQA |
| $\epsilon$ | Acc(%) | IMG | Acc(%) | Acc(%) | Acc(%) | IMG | Acc(%) | Acc(%) | Acc(%) | IMG | Acc(%) | Acc(%) |
| 1 | 23.30 | 21.50 | 1.13 | 0.00 | 81.54 | 72.51 | 34.52 | 38.61 | **84.20** | **78.43** | **34.74** | **39.06** |
| 3 | 21.50 | 19.90 | 2.82 | 0.00 | 78.80 | 70.59 | **35.64** | 39.11 | **82.80** | **75.98** | 35.17 | **39.65** |
| 8 | 21.50 | 19.90 | 1.31 | 0.00 | 82.52 | 74.00 | 35.60 | 39.16 | **85.10** | **76.47** | **35.71** | **39.78** |
| $\infty$ | 22.16 | 0.98 | 0.95 | 0.00 | 81.10 | 73.53 | 34.89 | 38.92 | **84.60** | **79.41** | **35.53** | **39.06** |

isolate the impact of DP fine-tuning methods, we do not perform additional instruction tuning stages beyond the initial pre-training of LLAVA. Parameter-efficient fine-tuning is achieved using LoRA Hu et al. (2022) (rank $r = 128$, scaling $\alpha = 256$) with batch size 12. The reported batch size ($B = 12$) serves as a user-facing hyperparameter to define the sampling rate $q = B/N$, where $N$ is the total number of training samples. Our privacy accounting rigorously uses this sampling rate with the Rényi Differential Privacy (RDP) Accountant, which correctly handles the mechanics of Poisson sampling.. All models are trained on the train set using the Adam optimizer Kingma & Ba (2014) with a learning rate of 2e-4 for 1 epoch. We use 4 A100 40G GPUs for training.

**DP Implementation.** We guarantee $(\epsilon, \delta)$-DP via the Gaussian Mechanism Privacy loss is tracked using Rényi Differential Privacy(RDP) Mironov (2017). We set $\delta$ close to the inverse dataset size ($1/N$) and evaluate across strict to mild privacy budgets: $\epsilon \in \{1, 3, 8\}$. Per-sample gradients are clipped at a maximum $L_2$ norm of $C = 1.0$.

**Baselines.** Our Dual-Priv Pruning method is compared against: DP-SGD Abadi et al. (2016): The standard baseline for DP fine-tuning, applying Gaussian noise to the averaged clipped gradients of all trainable parameters. DPZO Tang et al. (2024): A representative zeroth-order DP optimization method, included to assess alternatives that avoid direct gradient computation. Detailed for baselines are in Appendix G.

**Dual-Priv Pruning Configuration.** Mechanism 1 (Section 4.1) retains $K = 191$ attention-selected visual tokens plus [CLS] and 30 fused token (40% of total). Mechanism 2 (Section 4.2) employs gradient-update pruning by selecting parameter blocks for update if their noisy gradient norms are among the **top 80%** of all block norms (Eq. (10)).

## 5.2 PERFORMANCE ON STANDARD BENCHMARKS

The performance comparison on standard benchmarks is presented in Table 1. Our proposed Dual-Priv method consistently outperforms both DP-SGD and DP-ZO across all tested settings. The advantage is particularly pronounced on ScienceQA, where at $\epsilon = 3$, our method achieves an accuracy of **82.80%**, significantly surpassing DP-SGD (78.80%). This demonstrates our approach's superior ability to preserve essential visual information through token and gradient pruning. While DP-ZO's performance is non-competitive due to convergence issues, our method maintains a consistent edge over the strong DP-SGD baseline.

We observe a non-monotonic relationship between utility and the privacy budget $\epsilon$ in some individual cases, a known phenomenon in DP fine-tuning of large models where noise can act as a regularizer against overfitting Liu et al. (2025a; 2024). However, despite these local fluctuations, the *average* performance across all our benchmarks, summarized in Table 2, already conforms to the expected global privacy-utility trade-off. It shows that, on average,

Table 2: Average accuracy (%) on visual tasks, showing the global privacy-utility trend.

| Method | $\epsilon = 1$ | $\epsilon = 3$ | $\epsilon = 8$ | $\epsilon = \infty$ |
|---|---|---|---|---|
| DP-SGD | 48.54 | 48.44 | 49.11 | 49.58 |
| Dual-Priv | **50.74** | **50.26** | **50.65** | **51.33** |

higher privacy budgets generally lead to higher utility for both methods. To rigorously and definitively demonstrate this trade-off at a granular level, we performed an extended analysis under much stricter privacy constraints ($\epsilon \in \{0.5, 0.1, 0.05\}$). These results, detailed in **Appendix N**, show a clear and consistent trend where utility systematically declines as privacy becomes stricter. This confirms the robustness of our method and its adherence to the fundamental trade-off, especially in the challenging low-$\epsilon$ regime where our performance margin over baselines remains significant.

Table 3: Comparison on PathVQA and VQA-RAD under different DP budgets (ours on the right).

| $\epsilon$ | DPZO | | | | DP-SGD | | | | Ours (Dual-Priv) | | | |
|---|---|---|---|---|---|---|---|---|---|---|---|---|
| | PathVQA | | | VQA-RAD | PathVQA | | | VQA-RAD | PathVQA | | | VQA-RAD |
| | BLUE | EXT | F1 | Acc(%) | BLUE | EXT | F1 | Acc(%) | BLUE | EXT | F1 | Acc(%) |
| 1 | 0.6534 | 0.0301 | 0.0592 | 0.0 | 0.7222 | 0.3732 | 0.3675 | 47.3 | **0.7385** | **0.3840** | **0.3792** | **48.6** |
| 3 | 0.6534 | 0.0301 | 0.0592 | 0.0 | 0.7257 | 0.3712 | 0.3653 | 48.1 | **0.7263** | **0.3738** | **0.3701** | **48.8** |
| 8 | 0.6534 | 0.0301 | 0.0592 | 0.0 | 0.7140 | 0.3683 | 0.3635 | 46.8 | **0.7195** | **0.3763** | **0.3713** | **49.0** |

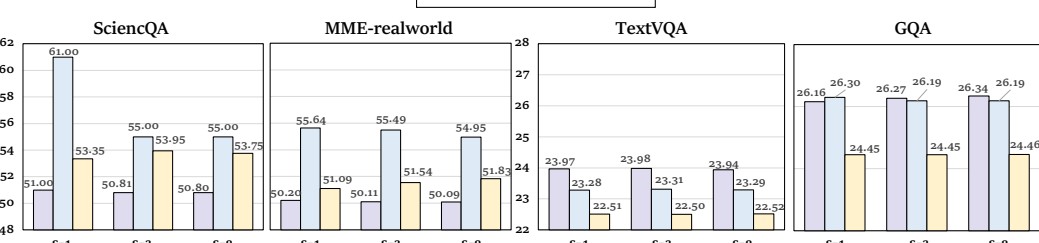

Figure 2: Average GPU memory consumption (in GB) during fine-tuning for DPZO, DP-SGD, and our Dual-Priv Pruning across four datasets: ScienceQA, MME-RealWorld (evaluated on 4xA100 40G GPUs), and TextVQA, GQA (evaluated on a single H20 96GB GPU). Experiments were conducted with varying privacy budgets ($\epsilon \in \{1, 3, 8\}$). Lower bars indicate greater memory efficiency.

## 5.3 PERFORMANCE ON MEDICAL VISUAL TASKS

To further assess applicability in privacy-sensitive domains, we evaluated performance on PathVQA He et al. (2020) (pathology) and VQA-RAD Lau et al. (2018) (radiology). Table 3 presents a detailed comparison of performance under different privacy budget. Our method, consistently outperformed DP-SGD across all the metrics, particularly under stricter privacy budgets. For $\epsilon = 1$: on PathVQA, our approach achieved scores of $0.74$(BLUE), $0.38$ (EXT), and $0.38$ (F1), compared to DP-SGD's $0.72$, $0.37$, and $0.37$, respectively. On VQA-RAD, our method achieved an accuracy of $48.60\%$, surpassing DP-SGD ($47.30\%$). The DPZO baseline performed poorly on both medical datasets. These consistent gains underscore the potential of Dual-Priv Pruning for tuning MLLMs on sensitive medical data while effectively balancing privacy and utility.

## 5.4 PERFORMANCE ON HIGH RESOLUTION VISUAL TASKS

We evaluate our method on the MME-RealWorld benchmark Zhang et al. (2024a) to test its efficacy on tasks requiring fine-grained visual perception, a known challenge for DP methods. After DP-finetuning on the training set, models are evaluated on the benchmark's lite version. As shown in Table 4, Dual-Priv demonstrates a decisive advantage over both baselines.

Table 4: Accuracy (%) on the MME-RealWorld Benchmark (Lite version evaluation, BS=12).

| Method | $\epsilon = 1$ | $\epsilon = 3$ | $\epsilon = 8$ | $\epsilon = \infty$ |
|---|---|---|---|---|
| DPZO | 0.89 | 19.80 | 6.33 | 22.67 |
| DP-SGD | 35.44 | 44.03 | 42.17 | 44.50 |
| Ours (Dual-Priv) | **43.98** | **45.34** | **44.40** | 42.16 |

Under the strict $\epsilon = 1$ constraint, our method reaches **43.98%** accuracy, significantly outperforming DP-SGD's 35.44%. This robust performance gain highlights Dual-Priv's effectiveness in preserving complex reasoning abilities despite DP noise, suggesting its strong potential for deploying privacy-preserving MLLMs in real-world applications

## 5.5 COMPUTATIONAL EFFICIENCY ANALYSIS

Figure 2 illustrates the average GPU memory usage during fine-tuning for our method compared to the baselines. Across the evaluated datasets, scienceqa on 4 A100s, Dual-Priv Pruning achieves an average **reduction in average GPU memory usage of approximately 14.34%**. Although DPZO slightly reduces 1.74% GPU memory compared with our approach. ( It costs 16.7% more time per training step and causes a 56.3% performance loss ). But during tested in H20, our method achieve the lowest consumption of GPU memory. This highlights Dual-Priv Pruning's strength in achieving a favorable balance between model performance and robust computational efficiency, thereby making DP fine-tuning for MLLMs more practical. Dual-Priv enables the DP fine-tuning of MLLMs with more constrained resources.

## 5.6 ABLATION STUDY

Ablation results on ScienceQA ($\epsilon = 1$) are in Table 5. The "w/o Fusion Noise" setting (omitting input-level noise) reduces performance to 83.50/76.47 (ACC/IMG) versus the Full Method's 84.20/78.43, indicating that preconditioning inputs with DP-consistent noise provides an auxiliary performance benefit without adding a tunable hyperparameter for this noise. Removing Mechanism 1 (token pruning; Mechanism 2 Only) lowers accuracy to 83.00/76.96 and eliminates the computational-efficiency benefits. Replacing Mechanism 2's selective update with uniform DP-SGD noise further degrades performance to 82.80/74.51, confirming the effectiveness of our adaptive update strategy. These findings demonstrate that both Mechanism 1 and Mechanism 2 are crucial components to the overall performance of dual-private pruning.

Table 5: Ablation on ScienceQA.

| Configuration | ACC | IMG |
|---|---|---|
| Full Method | **84.20** | **78.43** |
| w/o Fusion Noise | 83.50 | 76.47 |
| Mechanism 2 Only | 83.00 | 76.96 |
| Mechanism 1 + Uniform DP | 82.80 | 74.51 |

### 5.7 IMPACTS OF PRUNING RATIOS

We examine the impact of different pruning ratios within the Dual-Priv Pruning framework on the ScienceQA dataset ($\epsilon = 1$). Figure 3 (a) shows how the **gradient-update pruning ratio** relates to overall accuracy (ACC): ACC peaks at **84.20** when updating the top 80% of blocks; updating all blocks lowers ACC to 82.80, and lower ratios (60%, 50%, 10%) yield 81.60, 82.00, and 81.10. Figure 3 (b) shows how the **visual token retention ratio** affects image-based accuracy (IMG): IMG peaks at **78.43%** with

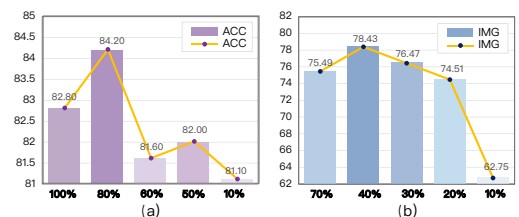

Figure 3: Pruning ratios impacts on ScienceQA ($\epsilon = 1$). (a) percentage of top K% gradient blocks updated (Mechanism 2). (b) percentage of visual tokens retained (Mechanism 1).

40% token retention; retaining more (70%) lowers IMG to 75.49, and retaining fewer reduces performance (76.47 at 30%, 74.51 at 20%), dropping sharply to 62.75 at 10%. These results highlight a trade-off in both pruning mechanisms: optimal performance retains sufficient signal (visual features or gradient updates) while pruning elements that might be redundant or overly affected by DP noise.

### 5.8 PERFORMANCE UNDER MEMBERSHIP INFERENCE ATTACKS

To further test the privacy protection capability of our approach, we validate the performance through membership inference attack. The latest MIA design for MLLM Li et al. (2024b) was adopted as the evaluation pipeline. Models were DP-finetuned on privacy sensitive medical image caption dataset **ROCOV2**Rückert et al. (2024) with the batchsize of 12 following the standard setup (Section 5.1). Extensive experiments demonstrate that our work outperform both DPZO and DPSGD across metrics include AUC and ACC, especially in protecting visual information as it benefit from adding heuristic fusion noise in Mechanism 1. As shown in Figure 4, attacks on our model yield the lowest AUC under almost every Rényi-entropy order and top-entropy selection percentage, meaning that under the same attack pipeline Dual-Priv Pruning makes distinguishing a database member least likely compared with other

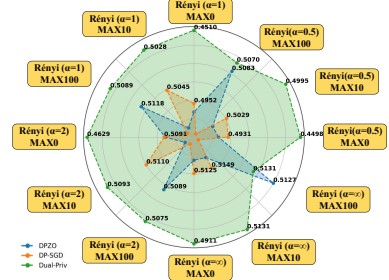

Figure 4: Radar chart of AUC under varying Rényi entropy orders and top entropy percentages. Metrics use strict privacy budget ($\epsilon = 1$). Distribution places smaller values near edges.

methods. Further data suggest that our method has a strong ability to prevent the MLLM from assigning a higher "membership score" to a randomly chosen member than to a randomly chosen non-member, bringing the membership-inference attack close to random guessing; see Appendix I for details.

### 6 CONCLUSION

In this work, we introduced **Dual-Priv Pruning**, the first framework for efficient differential private fine-tuning of Multimodal Large Language Models. Our approach combines *visual token pruning* with an input noise strategy aligned with DP noise intensity, and a *gradient-update pruning* mechanism. Extensive experiments demonstrate that Dual-Priv Pruning achieves a compelling privacy-utility trade-off, significantly reducing computational overhead while maintaining competitive performance, especially under stringent privacy budgets. This work represents a crucial first step towards practical privacy-preserving MLLM deployment.

## 7 ETHICS STATEMENT

This work introduces Dual-Priv Pruning, a framework that integrates visual token pruning and gradient-update pruning as complementary mechanisms for differentially private fine-tuning in MLLMs. We acknowledge the dual-use nature of security research and have taken deliberate steps to mitigate associated risks. All experiments are strictly confined to public benchmarks and open-source models, never involving deployed or proprietary systems, which ensures reproducibility while preventing real-world harm. This study did not involve new data collection, human subjects, or personally identifiable information, and complies with all dataset licenses. To prevent misuse, any released artifacts will be shared under a research-only license. We are committed to the responsible advancement of scientific knowledge, were mindful of our computational budget to limit environmental impact, and adhere to the ICLR Code of Ethics.

## 8 REPRODUCIBILITY STATEMENT

We ensure the reproducibility of our results by releasing the source code, baseline implementations, and all experimental scripts upon publication. Our evaluation is conducted entirely on publicly available benchmarks (ScienceQA, TextVQA, GQA, MME-RealWorld, PathVQA, and VQA-RAD) using accessible backbones (LLaVA-7B and Med-LLaVA). Detailed descriptions of datasets, model architectures, training strategies, hyperparameters, and privacy accounting are provided in Section 5.1 and the Appendix, enabling independent verification of our findings.

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

## A   KEY DIFFERENTIAL PRIVACY FACTS

The following facts elucidate key DP properties essential for MLLM fine-tuning:

**Fact A.1** (Sensitivity and the Gaussian Mechanism Dwork et al. (2014)). *To protect the output of a function $f$ using noise, we first need to leverage its $L_2$ sensitivity $\Delta f$. This measures the maximum possible change $\|f(\mathcal{D}) - f(\mathcal{D}')\|_2$ when the input dataset changes by one record. If $\Delta f$ is known (or bounded), the Gaussian Mechanism adds noise $\mathcal{N}(0, \sigma_{GM}^2 I)$ where the standard deviation $\sigma_{GM}$ is related to $\Delta f$ and depends on the desired single-step privacy $(\epsilon, \delta)$, calculated as:*

$$\sigma_{GM} \geq \frac{\Delta f \sqrt{2\ln(1.25/\delta)}}{\epsilon}. \tag{12}$$

**Fact A.2** (Privacy Accounting via RDP Mironov (2017)). *Privacy accounting methods are essential for tracking this privacy loss. Rényi Differential Privacy (RDP) Mironov (2017) is widely used for such accounting Abadi et al. (2016). The RDP accountant's practical role is to, given a target overall privacy budget $(\epsilon, \delta)$, total training steps $T$, and sampling rate $q$, compute the required per-step noise multiplier $\sigma$ ( 3.1.1) and suggest a clipping norm $C$ to meet the $(\epsilon, \delta)$-DP guarantee. Mathematical details are in Appendix C.*

## B   SENSITIVITY ANALYSIS FOR DP-SGD UNDER ADD-OR-REMOVE ADJACENCY

In DP-SGD (Definition 3.1.1), we apply the Gaussian Mechanism to the average of per-sample clipped gradients. The choice of adjacency definition for datasets $\mathcal{D}$ and $\mathcal{D}'$ (i.e., how they differ by "one record") impacts the sensitivity calculation. As stated in Definition 1, our work considers add-or-remove adjacency, where neighboring datasets differ by the addition or removal of a single image-text pair.

Consider the function $f(\mathcal{D}, \theta) = \sum_{x_i \in \mathcal{D}} \hat{g}_i(\theta)$, which is the sum of clipped gradients $\hat{g}_i$ for all samples $x_i$ in a dataset $\mathcal{D}$. Each per-sample clipped gradient satisfies $\|\hat{g}_i\|_2 \leq C$. If we consider two neighboring datasets $\mathcal{D}$ and $\mathcal{D}'$ where $\mathcal{D}' = \mathcal{D} \cup \{x^*\}$ (i.e., $x^*$ is added), then:

$$\|f(\mathcal{D}, \theta) - f(\mathcal{D}', \theta)\|_2 = \|\sum_{x_i \in \mathcal{D}} \hat{g}_i(\theta) - (\sum_{x_i \in \mathcal{D}} \hat{g}_i(\theta) + \hat{g}_{x^*}(\theta))\|_2 = \|-\hat{g}_{x^*}(\theta)\|_2 = \|\hat{g}_{x^*}(\theta)\|_2 \leq C.$$

Similarly, if $\mathcal{D}' = \mathcal{D} \setminus \{x^*\}$ (i.e., $x^*$ is removed), the difference is also bounded by $C$. Thus, the $L_2$ sensitivity of the sum of clipped gradients is $\Delta_2 f = C$.

In DP-SGD, we typically compute gradients over a minibatch $\xi_t$ of size $m$ sampled from the full dataset $\mathcal{D}_{\text{train}}$ (of size $N$) with sampling probability $q = m/N$. The noisy update is applied to the *average* of these clipped gradients: $\bar{g} = \frac{1}{m} \sum_{i \in \xi_t} \hat{g}_i$. For the per-iteration application of DP-SGD with minibatch sampling, the effective $L_2$ sensitivity of the quantity to which noise is added (the average gradient) is commonly taken as $\frac{C}{m}$ under the add-or-remove model when considering the privacy implications for each individual sample's contribution to this average Abadi et al. (2016); Dwork et al. (2014). More precisely, the clipping ensures that the maximum influence of any single user's data on the sum of gradients in a batch is $C$. When this sum is averaged over $m$ samples, the change due to one user's data (if that user were removed from the entire dataset) can be bounded appropriately, leading to the noise calibration based on $C$.

The noise added in DP-SGD (Eq. (2)) is $\mathcal{N}(0, \sigma^2 C^2 I_d / m^2)$. This formulation inherently uses $C$ as the sensitivity for the sum of gradients in the batch if we consider each sample's gradient to be a distinct quantity to be protected, and then this noise is scaled by $1/m$ due to averaging (equivalently, the sensitivity of the average is $C/m$). The critical point is that clipping each per-sample gradient to $C$ bounds its maximum possible contribution. The privacy analysis with subsampling (accounted for by the RDP accountant) then correctly tracks the privacy loss given this per-sample bound $C$.

Therefore, the clipping norm $C$ directly bounds the $L_2$ norm of each individual's contribution before aggregation and noise addition. The Gaussian Mechanism (Fact A.1) is then applied using this understanding, where the effective sensitivity for the noisy average gradient computation in DP-SGD is appropriately scaled by $C$.

# C RÉNYI DIFFERENTIAL PRIVACY (RDP) ACCOUNTING

RDP Mironov (2017) provides a way to track privacy loss using Rényi divergence of order $\alpha$, denoted $D_\alpha(P\|Q)$. An algorithm $\mathcal{A}$ is $(\alpha, \rho)$-RDP if for all neighboring datasets $\mathcal{D}, \mathcal{D}'$, $D_\alpha(\mathcal{A}(\mathcal{D})\|\mathcal{A}(\mathcal{D}')) \leq \rho$. Key properties include:

- **Composition:** If $\mathcal{A}_1$ is $(\alpha, \rho_1)$-RDP and $\mathcal{A}_2$ is $(\alpha, \rho_2)$-RDP, their composition $\mathcal{A}_2 \circ \mathcal{A}_1$ is $(\alpha, \rho_1 + \rho_2)$-RDP. This simplifies tracking loss over multiple steps.
- **Gaussian Mechanism RDP:** Adding $\mathcal{N}(0, \sigma_{GM}^2 I)$ noise to a function with $L_2$ sensitivity $\Delta f$ satisfies $(\alpha, \frac{\alpha(\Delta f)^2}{2\sigma_{GM}^2})$-RDP for any $\alpha > 1$.
- **Subsampling Amplification:** Sampling a minibatch with rate $q$ before applying a DP mechanism amplifies privacy. RDP provides tight bounds for this, especially for Poisson sampling Abadi et al. (2016) and uniform sampling without replacement.
- **Conversion to $(\epsilon, \delta)$-DP:** If an algorithm is $(\alpha, \rho)$-RDP for all $\alpha$ in some range, it satisfies $(\epsilon, \delta)$-DP where $\epsilon = \rho + \frac{\log(1/\delta)}{\alpha - 1}$. We typically optimize over $\alpha$ to find the tightest $\epsilon$ for a given $\delta$.

The privacy accountant takes $T, q$, the per-step RDP parameters (derived from the Gaussian mechanism using $C$ and $\sigma$), applies composition and subsampling rules to get the total RDP parameters $(\alpha, \rho_{total})$, and converts this to the final $(\epsilon, \delta)$. It can also work backwards: given target $(\epsilon, \delta), T, q$, find the required $\sigma$.

# D POST-PROCESSING PROPERTY OF DIFFERENTIAL PRIVACY

The post-processing property is a fundamental and powerful feature of differential privacy Dwork et al. (2014). It states that applying any arbitrary data-independent computation to the output of a differentially private algorithm does not compromise its privacy guarantee.

**Formal Statement:** Let $\mathcal{A} : \mathcal{D}^n \to \mathcal{R}$ be an $(\epsilon, \delta)$-differentially private algorithm, where $\mathcal{D}^n$ is the space of possible datasets and $\mathcal{R}$ is the output range. Let $f : \mathcal{R} \to \mathcal{R}'$ be any arbitrary randomized or deterministic function whose computation does not depend on the original private dataset $\mathcal{D}$ (it only takes the output of $\mathcal{A}$ as input). Then the composite algorithm $f \circ \mathcal{A}$ (which first runs $\mathcal{A}$ on the input dataset and then applies $f$ to the result) is also $(\epsilon, \delta)$-differentially private.

**Intuition:** The privacy guarantee provided by $\mathcal{A}$ ensures that its output $Y = \mathcal{A}(D)$ is already "privacy-safe" – observing $Y$ reveals limited information about any individual in $D$. The function $f$ only gets access to this already protected output $Y$. Since $f$ has no additional access to the original sensitive data $D$, it cannot "undo" the privacy protection or learn anything more about individuals in $D$ than what was already bounded by the $(\epsilon, \delta)$-DP guarantee of $\mathcal{A}$.

**Proof Sketch:** We want to show that for any neighboring datasets $D, D'$ and any set of outcomes $S' \subseteq \mathcal{R}'$:
$$\Pr[(f \circ \mathcal{A})(D) \in S'] \leq e^\epsilon \cdot \Pr[(f \circ \mathcal{A})(D') \in S'] + \delta$$
Let $Y = \mathcal{A}(D)$ and $Y' = \mathcal{A}(D')$. The event $(f \circ \mathcal{A})(D) \in S'$ means $f(Y) \in S'$. Let $S_f = \{y \in \mathcal{R} \mid f(y) \in S'\}$ be the set of outputs from $\mathcal{A}$ that $f$ maps into $S'$. Then, $\Pr[(f \circ \mathcal{A})(D) \in S'] = \Pr[Y \in S_f] = \Pr[\mathcal{A}(D) \in S_f]$. Similarly, $\Pr[(f \circ \mathcal{A})(D') \in S'] = \Pr[\mathcal{A}(D') \in S_f]$. Since $\mathcal{A}$ is $(\epsilon, \delta)$-DP and $S_f$ is a valid subset of its output range $\mathcal{R}$, we know from Definition 1:
$$\Pr[\mathcal{A}(D) \in S_f] \leq e^\epsilon \cdot \Pr[\mathcal{A}(D') \in S_f] + \delta$$
Substituting back, we get:
$$\Pr[(f \circ \mathcal{A})(D) \in S'] \leq e^\epsilon \cdot \Pr[(f \circ \mathcal{A})(D') \in S'] + \delta$$
This holds for any $S'$, proving that $f \circ \mathcal{A}$ is $(\epsilon, \delta)$-DP.

**Relevance to MLLM Fine-tuning:** In our context, the DP-SGD algorithm $\mathcal{A}$ takes the private dataset $\mathcal{D}_{\text{fine}}$ and produces the model parameters $\theta_{\text{fine}}$. The act of generating a prediction for a new input $x$, i.e., computing $M_{\theta_{\text{fine}}}(x)$, can be viewed as a post-processing function $f$ applied to $\theta_{\text{fine}}$. Therefore, the generated predictions inherit the same $(\epsilon, \delta)$-DP guarantee with respect to the fine-tuning dataset $\mathcal{D}_{\text{fine}}$.

# E    MOTIVATION FOR MECHANISM 1: VISUAL TOKEN PRUNING AND FUSION

This appendix details the motivation behind the visual input preprocessing performed in Mechanism 1 of our Dual-Priv Pruning method (Section 4). This stage operates *before* the formal Differentially Private (DP) fine-tuning in Mechanism 2 (§4.2) and is designed to address key challenges in applying DP to Multimodal Large Language Models (MLLMs). Specifically, it aims to reduce computation cost and potentially improve the utility outcome under DP constraints by modifying the visual token sequence.

## E.1    ADDRESSING COMPUTATION COST AND VISUAL REDUNDANCY

Fine-tuning MLLMs using DP-SGD can be computationally demanding, due to the high number of visual tokens ($n$) generated by the vision encoder. It has been observed that considerable redundancy exists within the visual tokens generated by Vision Transformers (ViTs), and not all tokens are equally important for downstream task performance Kong et al. (2022); Haurum et al. (2023). Building on the insight that attention scores often correlate with token importance Haurum et al. (2023), Mechanism 1 identifies and retains only the top-$K$ tokens receiving the highest aggregated attention from the [CLS] token. This selective pruning significantly shortens the sequence length processed in Mechanism 2, thereby directly reducing computation overhead. This strategy aligns with research exploring attention-based token pruning in ViTs Kong et al. (2022); Rao et al. (2021).

## E.2    PRESERVING CONTEXT VIA TOKEN FUSION

While pruning reduces costs, simply discarding less attended tokens might remove valuable contextual information. To mitigate this, Mechanism 1 adopts a fusion strategy inspired by techniques that aim to compress information from pruned parts of a network or input Wei et al. (2023). We merge the non-dominant tokens ($\mathcal{V}_{nd}$) into selected context tokens ($c$). This allows us to maintain a drastically reduced sequence length for efficiency while still incorporating a summarized representation of the less critical visual information, aiming for a better balance between computational savings and information preservation.

## E.3    HEURISTIC NOISE INJECTION: MOTIVATIONS AND POTENTIAL BENEFITS

The final step of Mechanism 1 introduces heuristic Gaussian noise to the fused context tokens ($c$) (Eq. equation 7). This deliberate noise injection is multifaceted, aiming to potentially enhance the subsequent DP fine-tuning process:

- **Regularization against DP Noise:** Adding noise is a known regularization technique (Bishop, 1995; Noh et al., 2017). Injecting noise specifically into the summarized, less critical token representation might act as **targeted input regularization**. This could potentially improve the model's robustness against the gradient perturbations inherent in the DP mechanism.

- **Encouraging Focus on Critical Tokens:** By introducing stochasticity primarily to the fused context token, the model might be implicitly encouraged during fine-tuning to rely more heavily on the stable, un-noised dominant tokens ($\mathcal{V}_d, v_{cls}$). This could help **preserve utility related to salient image features**.

- **Connection to Learning with Noise Priors:** Although mechanically different, this strategy shares a conceptual link with methods improving DP training by incorporating knowledge from noisy processes Tang et al. (2023a).Our direct noise injection might serve a similar purpose by **preconditioning the model with input stochasticity**, potentially enhancing its resilience to the noise required for the DP guarantee in Mechanism 2.

- **Conceptual Input-Level Obfuscation:** While not contributing to the formal DP guarantee, manipulating the representation of less critical tokens with heuristic noise offers a degree of **data obfuscation at the input level**. This might provide some practical hardening against certain inference attacks targeting those specific, less informative image regions.

It is crucial to emphasize that the noise added in Mechanism 24.1 ($\sigma_{fuse}^2$) is **heuristic**. It is not calibrated according to DP principles and serves as a hyperparameter tuned for its potential benefits to utility and robustness.

## F    Motivation for Mechanism 2 Gradient-Update Pruning

The post-noise adaptive update mechanism described in Section 4.2 is motivated by the goal of enhancing model utility under the constraints imposed by DP-SGD noise. Standard DP-SGD applies the noisy gradient $\tilde{g}$ (Eq. (9)) uniformly to all trainable parameters $\theta_{\text{train}}$. However, the added noise can significantly perturb or even dominate the true gradient signal, especially for parameter blocks where the original gradient magnitude was small or when operating under strict privacy budgets (requiring large $\sigma$). Applying updates based on such noise-dominated gradients might hinder convergence or lead to suboptimal performance.

Our strategy addresses this by analyzing the noisy gradient $\tilde{g}$ *after* the privacy-preserving noise has been added. By partitioning $\tilde{g}$ into logical blocks $\tilde{g}_j$ and examining their $L_2$ norms $N_j = \|\tilde{g}_j\|_2$, we attempt to identify blocks where the signal likely outweighs the noise. The assumption is that a relatively large norm $N_j$ suggests that the original aggregated gradient component $\hat{g}_j$ was sufficiently strong to persist despite the noise addition, thus indicating a more reliable update direction. Conversely, a small norm $N_j$ might indicate that the true signal was weak or was largely cancelled by the random noise vector.

The gating mechanism (Eq. (10)) leverages this analysis. By generating a mask $M$ that selectively allows updates only for blocks with high noisy-gradient norms (i.e., where $M_j = 1$), we filter out potentially detrimental updates arising from low-signal or noise-dominated gradient components. The final masked update (Eq. (11)) focuses the optimization process on parameter blocks associated with stronger, potentially more informative, noisy gradient signals. This aims to improve the effective signal-to-noise ratio of the updates applied to the model, potentially leading to better convergence, improved utility, and a more favorable privacy-utility trade-off for the given privacy budget $(\epsilon, \delta)$.

## G    Baseline Details

This section provides detailed descriptions, algorithms, and hyperparameter configurations for the baseline methods used in our comparative experiments.

### G.1    DP-SGD Baseline

We implement the standard Differentially Private Stochastic Gradient Descent (DP-SGD) algorithm Abadi et al. (2016), formally defined in 3.1.1. This method involves computing per-sample gradients, clipping their $L_2$ norms, averaging the clipped gradients, and adding calibrated Gaussian noise before updating the model parameters. It serves as the primary benchmark for differentially private optimization in deep learning. The hyperparameter configuration used for DP-SGD is detailed in Table 6.

### G.2    DPZO Baseline

DPZO (Differentially Private Zeroth-Order Optimization) Tang et al. (2024) is a gradient-free DP optimization method. It approximates the gradient direction using finite differences based on random perturbations and privatizes only a scalar value representing the estimated directional derivative (loss difference). This avoids the memory overhead associated with storing per-sample gradients, but often requires significantly more iterations for convergence compared to DP-SGD. Algorithm 3 outlines the core mechanism adapted from Tang et al. (2024). The specific configuration used in our experiments is presented in Table 7.

Table 6: Hyperparameter Configuration for DP-SGD Baseline.

| Parameter | Value |
|---|---|
| Base Model | LLAVA-7B Liu et al. (2023) |
| Fine-tuning Method | LoRA Hu et al. (2022) |
| LoRA Rank ($r$) | 128 |
| LoRA Alpha ($\alpha$) | 256 |
| Optimizer | Adam Kingma & Ba (2014) |
| Learning Rate | 2e-4 |
| Batch Size | 12 |
| Epochs | 1 |
| **DP Parameters** | |
| Clipping Norm ($C$) | 1.0 |
| Target $\delta$ | $\approx 1/N$ (Inverse dataset size) |
| Target $\epsilon$ Values | $\{1, 3, 8, \infty\}$ |
| Noise Multiplier ($\sigma$) | Calculated via RDP Mironov (2017) based on target $(\epsilon, \delta)$, $C$, $q$, and total steps. |

Table 7: Hyperparameter Configuration for DPZO Baseline.

| Parameter | Value |
|---|---|
| Base Model | LLAVA-7B Liu et al. (2023) |
| Fine-tuning Method | LoRA Hu et al. (2022) |
| LoRA Rank ($r$) | 128 |
| LoRA Alpha ($\alpha$) | 256 |
| Learning Rate ($\eta$) | 2e-4 |
| Perturbation Scale ($\phi$) | 0.15 |
| Batch Size | 12 |
| Epochs | 1 |
| **DP Parameters** | |
| Clipping Norm ($C_{ZO}$) | 1.0 |
| Target $\delta$ | $\approx 1/N$ |
| Target $\epsilon$ Values | $\{1, 3, 8, \infty\}$ |
| Noise Multiplier ($\sigma_{ZO}$) | Calculated via RDP accountant based on target $(\epsilon, \delta)$, $C_{ZO}$, $q = m/N$, $T$. |

## H DETAILED RESULTS ON MEDICAL DATASETS

This section provides the detailed performance results for the experiments on the PathVQA and VQA-RAD datasets, as referenced in Section 5.1. All experiments used a batch size (BS) of 12.

Table 8: Detailed performance on PathVQA (BS=12). Higher is better for BLUE, EXT, F1. Best DP results in **bold**.

| $\epsilon$ | Ours (Dual-Priv) | | | DPZO | | | DP-SGD | | |
|---|---|---|---|---|---|---|---|---|---|
| | BLUE | EXT | F1 | BLUE | EXT | F1 | BLUE | EXT | F1 |
| 1 | **0.7385** | **0.3840** | **0.3792** | 0.6534 | 0.0301 | 0.0592 | 0.7222 | 0.3732 | 0.3675 |
| 3 | **0.7263** | **0.3738** | **0.3701** | 0.6534 | 0.0301 | 0.0592 | 0.7257 | 0.3712 | 0.3653 |
| 8 | **0.7195** | **0.3763** | **0.3713** | 0.6534 | 0.0301 | 0.0592 | 0.7140 | 0.3683 | 0.3635 |
| $\infty$ | 0.7430 | 0.3880 | 0.3841 | 0.6534 | 0.0301 | 0.0592 | 0.7182 | **0.3927** | **0.3879** |

## I ADDITIONAL DETAILS ON MEMBERSHIP INFERENCE ATTACK

This section provides the additional details for the experiments with membership inference attack, as referenced in Section 5.8. All experiments used a batch size(BS) of 12. We randomly sample 6,000 image-text pairs from the ROCOV2 dataset for evaluation and randomly sampled 3000 image-text pairs as the member dataset for training. To fit the LLaVA-VQA formulation, we randomly use these

Table 9: Detailed accuracy (%) on VQA-RAD (BS=12). Higher is better. Best DP result in **bold**.

| $\epsilon$ | Ours (Dual-Priv) | DPZO | DP-SGD |
|---|---|---|---|
| 1 | **48.6** | 0.0 | 47.3 |
| 3 | **48.8** | 0.0 | 48.1 |
| 8 | **49.0** | 0.0 | 46.8 |
| $\infty$ | 47.9 | 0.0 | **48.3** |

prompts:"Please describe the image in detail.", "What is shown in this medical image?", "Describe the contents of this image.", "What does this image depict?", "Provide a detailed description of this image.", "Please analyze this medical image.", "Describe the medical image in detail.", "Describe the condition depicted in the image.", "Please provide a caption for this image."

## J    LIMITATIONS

Our study demonstrates the effectiveness of Dual-Priv Pruning for DP fine-tuning MLLMs. While our evaluations on a 7B MLLM are thorough, extending the assessment to MLLMs of substantially different scales would provide a broader understanding of the approach's scalability.

## K    BROADER IMPACTS

The development of Dual-Priv Pruning contributes to the critical area of privacy-preserving machine learning, particularly for Multimodal Large Language Models (MLLMs). The primary societal benefit lies in its potential to significantly enhance data privacy when fine-tuning MLLMs on sensitive datasets. By integrating Differential Privacy (DP) with improved efficiency and utility, our work can empower the responsible use of MLLMs in domains handling personal information, such as healthcare or finance, thereby protecting individuals from data leakage. This advancement may also lower barriers to adopting privacy-enhancing technologies, encouraging a broader shift towards responsible AI practices and facilitating research on valuable sensitive datasets that might otherwise remain underutilized due to privacy risks. Ultimately, robust privacy measures like those explored can foster greater public trust in AI systems, which is vital for their ethical and successful integration into society.

However, it is important to consider the broader context. While DP offers strong mathematical privacy guarantees, these are contingent upon correct implementation and careful parameter selection, and they address specific threats related to individual data contributions rather than all conceivable privacy concerns. A nuanced understanding is crucial to avoid a false sense of absolute security. The inherent trade-off between privacy protection and model utility, though mitigated by our approach, persists; in certain high-stakes applications, even minor performance degradation due to DP noise could have notable implications if not carefully weighed. Furthermore, the expertise required to effectively implement and tune DP mechanisms remains a consideration for broader accessibility. While our method focuses on the privacy of training data, the underlying MLLM technology itself, regardless of how it's fine-tuned, could still be subject to misuse if its outputs are leveraged for unintended or harmful purposes.

Our research is a step towards more responsible AI development. We believe continued efforts in the community are essential to further refine the balance between privacy and utility, enhance the usability of DP tools, and promote comprehensive education on both the capabilities and limitations of such privacy-enhancing technologies. Addressing fairness and bias within DP-trained models also remains an important ongoing pursuit. This work is presented as foundational research to advance privacy in MLLM fine-tuning, with the anticipation that its net impact will be positive by enabling more secure and trustworthy AI applications.

## L   ALGORITHM FOR BASELINES AND DUAL-PRIV PRUNING

Algorithm 1 provides the detailed step-by-step procedure for the Stage 2 DP fine-tuning process described in Section 4.2 of the main paper. While Algorithm 2 outlines the standard DP-SGD baseline, and Algorithm 3 details the DPZO baseline.

---

**Algorithm 1** Dual-Priv Pruning: Mechanism 2 (DP Fine-tuning with Gradient-Update Pruning)

---

**Require:** Initial trainable parameters $\theta_{\text{train}_0}$, dataset $D = \{(\mathcal{V}'_i, \mathcal{T}_i)\}_{i=1}^N$ (with pre-processed visual inputs $\mathcal{V}'_i$), learning rate schedule $\eta_t$, gradient clipping norm $C$, noise multiplier $\sigma$ (derived from target $\epsilon, \delta$), batch size $m$, total training steps $T$, number of logical parameter blocks $J$ in $\theta_{\text{train}}$, top-K percentage $P_K$.

1: **for** $t = 1, \ldots, T$ **do**
2:     Sample minibatch $\xi_t = \{(\mathcal{V}'_k, \mathcal{T}_k)\}_{k=1}^m \subset D$ of size $m$.
3:     Initialize list of per-sample gradients $G_{list} = []$.
4:     **for** each sample $(\mathcal{V}'_k, \mathcal{T}_k) \in \xi_t$ **do**
5:         Compute gradient $g_k \leftarrow \nabla_{\theta_{\text{train}_{t-1}}} \mathcal{L}(\theta_{\text{train}_{t-1}}; (\mathcal{V}'_k, \mathcal{T}_k))$.
6:         Clip gradient: $\hat{g}_k \leftarrow g_k / \max(1, \|g_k\|_2 / C)$.
7:         Append $\hat{g}_k$ to $G_{list}$.
8:     **end for**
9:     Aggregate clipped gradients: $\bar{\hat{g}} \leftarrow \frac{1}{m} \sum_{\hat{g}_k \in G_{list}} \hat{g}_k$.
10:     Add Gaussian noise: $\tilde{g} \leftarrow \bar{\hat{g}} + \mathcal{N}\left(0, \frac{\sigma^2 C^2}{m^2} I_{d_{\text{train}}}\right)$.
11:     Partition $\tilde{g}$ into $J$ components $\{\tilde{g}_1, \ldots, \tilde{g}_J\}$ corresponding to logical parameter blocks.
12:     Compute $L_2$ norms for each block: $N_j \leftarrow \|\tilde{g}_j\|_2$ for $j = 1, \ldots, J$.
13:     Initialize mask $M$ as a zero tensor with the same block structure as $\theta_{\text{train}}$.
14:     Let $K_{\text{count}} \leftarrow \lceil (P_K/100) \cdot J \rceil$.
15:     Let $\mathcal{S}_{\text{top\_indices}}$ be the set of indices of the $K_{\text{count}}$ blocks with the largest norms $N_j$.
16:     **for** each block index $j \in \mathcal{S}_{\text{top\_indices}}$ **do**
17:         Set corresponding part of mask $M_j \leftarrow \mathbf{1}$ (vector/matrix of ones for block $j$).
18:     **end for**
19:     Update parameters: $\theta_{\text{train}_t} \leftarrow \theta_{\text{train}_{t-1}} - \eta_t \cdot (M \odot \tilde{g})$.
20: **end for**
21: **return** $\theta_{\text{train}_T}$.

---

**Algorithm 2** Differentially Private Stochastic Gradient Descent (DP-SGD, adapted from Abadi et al. (2016))

---

**Require:** Initial model parameters $\theta_0$, dataset $D = \{(\mathcal{I}_i, \mathcal{T}_i)\}_{i=1}^N$ (or generic $x_i$), learning rate $\eta_t$, clipping norm $C$, noise multiplier $\sigma$, batch size $m$, total steps $T$.

1: **for** $t = 1, \ldots, T$ **do**
2:     Sample minibatch $\xi_t = \{x_k\}_{k=1}^m \subset D$ of size $m$.
3:     Initialize list of per-sample gradients $G_{list} = []$.
4:     **for** each sample $x_k \in \xi_t$ **do**
5:         Compute gradient $g_k \leftarrow \nabla_{\theta_{t-1}} \mathcal{L}(\theta_{t-1}; x_k)$.
6:         Clip gradient: $\hat{g}_k \leftarrow g_k / \max(1, \|g_k\|_2 / C)$.
7:         Append $\hat{g}_k$ to $G_{list}$.
8:     **end for**
9:     Aggregate clipped gradients: $\bar{\hat{g}} \leftarrow \frac{1}{m} \sum_{\hat{g}_k \in G_{list}} \hat{g}_k$.
10:     Add Gaussian noise: $\tilde{g} \leftarrow \bar{\hat{g}} + \mathcal{N}\left(0, \frac{\sigma^2 C^2}{m^2} I_d\right)$.
11:     Update parameters: $\theta_t \leftarrow \theta_{t-1} - \eta_t \cdot \tilde{g}$.
12: **end for**
13: **return** $\theta_T$.

---

---

**Algorithm 3** DPZO Core Mechanism (Simplified, adapted from Tang et al. (2024))

---

**Require:** Model parameters $\theta$, dataset $D$, learning rate $\eta$, perturbation scale $\phi$, clipping threshold $C_{ZO}$, noise scale $\sigma_{ZO}$, batch size $m$, total steps $T$.

1: **for** $t = 1, \dots, T$ **do**
2:      Sample batch $B \subset D$.
3:      Sample random direction $z_t \sim \mathcal{N}(0, I_d)$.
4:      Set $\theta^+ \leftarrow \theta_{t-1} + \phi z_t$, $\theta^- \leftarrow \theta_{t-1} - \phi z_t$.
5:      Initialize loss differences list $L_{\text{diff}} = []$.
6:      **for** each sample $(\mathcal{I}_i, \mathcal{T}_i) \in B$ **do**
7:          Compute $l_i = \mathcal{L}(\theta^+; (\mathcal{I}_i, \mathcal{T}_i)) - \mathcal{L}(\theta^-; (\mathcal{I}_i, \mathcal{T}_i))$.
8:          Clip difference: $\hat{l}_i \leftarrow \max(-C_{ZO}, \min(l_i, C_{ZO}))$.
9:          Append $\hat{l}_i$ to $L_{\text{diff}}$.
10:      **end for**
11:      Aggregate clipped differences: $\bar{l} \leftarrow \frac{1}{|B|} \sum_{\hat{l}_i \in L_{\text{diff}}} \hat{l}_i$.
12:      Add noise to privatize the average difference: $s \leftarrow \bar{l} + \mathcal{N}(0, \sigma_{ZO}^2 C_{ZO}^2 / |B|^2)$.
13:      Update parameters: $\theta_t \leftarrow \theta_{t-1} - \eta \cdot s \cdot z_t / (2\phi)$.
14: **end for**
15: **return** $\theta_T$.

---

## M   WHY DUAL-PRIV PRUNING IMPROVES UTILITY

**DP noise as implicit regularization.** For over-parameterized pretrained models, the Gaussian noise in DP-SGD can limit memorization and act as an effective regularizer, which explains occasional non-monotonicity of utility w.r.t. $\epsilon$ at medium privacy levels (Liu et al., 2025b).

**Gradient-update pruning improves signal-to-noise.** After privatization in DP-SGD (Abadi et al., 2016), we update only the top-$K\%$ parameter blocks ranked by noisy-gradient norm. This DP-preserving post-processing reduces the *effective* update dimensionality from $d$ to $k \ll d$, aligning with analyses of sparsified DP optimization where the noise term scales with $\sqrt{k}$ rather than $\sqrt{d}$ (Zhu & Blaschko, 2023). Practically, this concentrates updates on LoRA-style modules that carry stronger task signal, yielding larger gains for large pretrained models and tighter privacy budgets. Our method is not superior to DP-SGD in all cases but is expected to show the most significant advantages under the following conditions:

When the model is a large, high-dimensional, and parameter-redundant pre-trained model: MLLMs are a typical example. Fine-tuning only requires updating a sparse subset of parameters. Applying DP-SGD to all parameters is inefficient and adds noise to already well-learned weights. When the privacy budget is strict (small $\epsilon$): In this high-noise environment, the SNR of standard DP-SGD drops sharply, and the advantages of our pruning method become more pronounced. Standard DP-SGD may be a better or more robust choice in the following situations:

When training a model from scratch: Gradient signals are likely dense across the parameter space, and pruning may discard useful information. When the model has a small number of parameters and no redundancy: In this case, $k$ close to $d$ , the theoretical benefit from pruning is small, and it might even harm performance by mistakenly pruning critical small gradient updates.

**Visual token pruning concentrates supervision; fusion noise aligns robustness.** Attention-based pruning removes redundant visual tokens and shortens sequences, focusing gradients on salient content. The small Gaussian *fusion* noise on compressed context tokens serves as input-level stochastic regularization whose scale is aligned with the downstream DP noise, echoing results showing that learning from noise-based priors can improve robustness under DP (Tang et al., 2023b). Empirically this helps under DP, but it is not a general booster for non-private fine-tuning.

**When to expect improvements.** The benefits are most pronounced with large, overparameterized pretrained models, moderate-to-tight privacy budgets, and vision-heavy inputs with redundancy. Under extremely strict privacy, utility inevitably drops, yet the approach remains more robust than plain DP-SGD and zeroth-order baselines in our experiments.

Table 10: Performance under extremely strict privacy budgets (lower $\epsilon$). Best per row is **bold**.

| Dataset | Metric | $\epsilon$ | Dual-Priv (Ours) | DP-SGD | DP-ZO |
|---------|--------|------------|------------------|--------|-------|
| ScienceQA | IMG | 0.5 | **76.98** | 75.00 | 10.29 |
|  |  | 0.1 | **61.23** | 60.08 | 1.96 |
|  |  | 0.05 | **43.14** | 42.90 | 0.00 |
| GQA | ACC | 0.5 | **39.20** | 38.81 | 0.00 |
|  |  | 0.1 | **38.28** | 37.90 | 0.00 |
|  |  | 0.05 | **37.79** | 36.70 | 0.00 |
| TextVQA | ACC | 0.5 | **34.76** | 33.84 | 0.34 |
|  |  | 0.1 | **32.68** | 32.10 | 0.16 |
|  |  | 0.05 | **31.27** | 30.17 | 0.12 |
| MME-RW | ACC | 0.5 | **42.68** | 28.04 | 0.78 |
|  |  | 0.1 | **28.50** | 25.79 | 0.67 |
|  |  | 0.05 | **24.65** | 24.14 | 0.04 |

## N  UTILITY BEHAVIOR UNDER TIGHT PRIVACY CONSTRAINTS

We further evaluate with very tight privacy budgets $\epsilon \in \{0.5, 0.1, 0.05\}$ (with $\delta \approx 1/N$ and the same backbone/LoRA/optimizer settings as in the main experiments). Results are summarized in Table 10.

**Emergence of the expected privacy–utility trade-off.**  When the privacy budget becomes very tight ($\epsilon \leq 1$), the classic trade-off appears consistently. For instance, on ScienceQA the IMG metric decreases from 78.43 at $\epsilon=1$ to 43.14 at $\epsilon=0.05$, indicating that the regularization benefit of DP noise at medium privacy levels is eventually outweighed as noise grows.

**Robustness of the proposed method.**  Across ScienceQA, TextVQA, and MME-RealWorld, Dual-Priv maintains clear advantages over DP-SGD and large margins over DP-ZO, which largely collapses at low $\epsilon$. On GQA, the gap to DP-SGD is small (DP-SGD is slightly higher at $\epsilon \in \{0.1, 0.05\}$), but performance remains comparable. We attribute the robustness to (i) concentrating supervision via visual-token pruning and (ii) post-noise gradient-update pruning, which restricts updates to signal-dominant blocks and reduces effective noise exposure.

## O  TEXT-AGNOSTIC VISUAL TOKEN PRUNING VIA [CLS] ATTENTION

**Motivation: quality over quantity.**  A growing body of work shows that many vision tokens in VLMs are redundant or even harmful when their semantics are misaligned; improving *which* tokens are kept matters more than keeping *more* tokens (Yang et al., 2024; Vasu et al., 2024; Shang et al., 2024; Zhang et al., 2024b). This motivates a saliency signal that is (i) model-internal, (ii) stable across prompts, and (iii) cheap to compute.

**Why use [CLS] attention.**  In ViT-style encoders, the [CLS] token aggregates global evidence; attention from [CLS] to patches highlights positions that contribute most to the image-level representation. Ranking tokens by averaged $[\text{CLS} \rightarrow \text{patch}]$ attention therefore targets the model's own notion of saliency and alleviates "feature misalignment," where object evidence is pooled into proxy tokens at non-intuitive locations (Yang et al., 2024). This aligns with attention-guided token reduction that consistently improves utility in prior work (Vasu et al., 2024; Shang et al., 2024; Zhang et al., 2024b).

**Why text-agnostic.**  Question-conditioned pruning can overfit to a single prompt and incurs extra cross-modal passes. Our design computes [CLS]-based saliency *once* from the vision encoder and reuses it for all instructions, making the pruning decision instruction-invariant (text-agnostic) and lower-variance. Crucially, because it depends only on visual features, Mechanism 1 (§4.1) is pure

pre-processing and does not consume privacy budget, while still shortening sequences before DP optimization in Mechanism 2.

**Retaining global context.** To avoid over-pruning fine-grained/background cues, non-dominant tokens are fused into a small set of context tokens with light Gaussian perturbation (§4.1). Ablations show that retaining ~40% tokens with fusion yields the best IMG while reducing compute (Fig. 3).

**Takeaway.** [CLS]-attention provides a prompt-invariant, compute-efficient, and privacy-friendly saliency signal that matches how ViTs aggregate information, complements evidence that token *quality* drives VLM utility (Yang et al., 2024; Vasu et al., 2024; Shang et al., 2024; Zhang et al., 2024b), and integrates naturally with our DP fine-tuning pipeline.

## P USE OF LARGE LANGUAGE MODELS (LLMs)

We used a large language model solely for language editing (grammar and fluency). It was not involved in research ideation, experimental design, implementation, data analysis, or citation selection; all technical content was authored and verified by the human authors.

