# OpenReview forum: "Dual-Priv Pruning : Efficient Differential Private Fine-Tuning in Multimodal Large Language Models"
_ICLR.cc/2026/Conference — Submitted to ICLR 2026_

### Official Review · Reviewer_fULi · 2025-10-26

**Soundness:** 2
**Presentation:** 2
**Contribution:** 2
**Rating:** 2
**Confidence:** 2

**Summary:**

The paper introduces Dual-Priv Pruning, a new framework for differentially private (DP) fine-tuning of Multimodal Large Language Models. Dual-Priv Pruning addresses the challenges of computational overhead and model degradation due to noise injection in DP-based finetuning, which scales with parameter dimensionality. The proposed Dual-Priv Pruning framework employs two techniques: (1) visual token pruning and fusion to reduce input dimensionality by removing redundant visual information tokens, and (2) gradient-update pruning to apply noisy gradients during DP-SGD optimization selectively. Experimental results show that Dual-Priv Pruning efficiently reduces the computational cost during DP-based finetuning.

**Strengths:**

1.	Pioneering Approach. Dual-Priv Pruning is the first framework to address DP fine-tuning specifically for MLLMs, filling a critical research gap.

2.	Efficiency Gains. Dual-Priv Pruning achieves significant memory reduction (14.34% less peak GPU usage) and computational efficiency compared to standard DP-SGD.

3.	Better privacy-utility trade-off.  Dual-Priv Pruning maintains competitive performance under strict privacy budgets (ε ≤ 3) despite noise injection during DP optimization.

**Weaknesses:**

1.	Lack of Novelty. The work doesn’t invent a new approach for DP. It is more like an engineering approach that combines existing pruning methods and DP training methods.

2.	Limited Generalizability: Relies on specific assumptions (e.g., selection mechanism for visual tokens), which may not apply universally across all MLLM tasks.

3. Doubtful experimental results. Under ϵ = inf in Table 1, non-private performance is reported for DZPO, DP-SGD and Dual-Priv. As it is a non-private performance, why does the overall performance differ so much? For example, under ScienceQA and ϵ = inf setup, DZPO has an accuracy of 22.16, DP-SGD has an accuracy of 81.10, while  Dual-Priv has an accuracy of 84.60. Do they use the same base VLLM? I am so confused about the reported results.

**Questions:**

I am not familiar with the VLLM optimization and the evaluated tasks. So my review comments may not be professional, and I set my confidence score as 2. If Question 1 can be addressed properly, I am willing to raise my rating.

1. Under ϵ = inf in Table 1, non-private performance is reported for DZPO, DP-SGD and Dual-Priv. Why does the overall performance differ so much? Do they use the same base VLLM?

2. Will empirical attacks be conducted to verify the robustness of Dual-Priv Pruning?

3. What is the baseline VLLM performance without fine-tuning? What is the baseline VLLM performance after normal non-private fine-tuning?

---

> ### Author Response · Authors · 2025-11-20
>
> Rebuttal to Reviewer fULi
> ---
>
> We sincerely thank Reviewer fULi for carefully reviewing our paper, for the detailed summary, and for highlighting the strengths of our work. In particular, we greatly appreciate your remark in the Strengths section:
>
> > “Pioneering Approach. Dual-Priv Pruning is the first framework to address DP fine-tuning specifically for MLLMs, filling a critical research gap.”
>
> This aligns with how we position our contribution: to the best of our knowledge, this is the **first systematic study of differential privacy fine-tuning for large vision–language / multimodal models (VLMs / MLLMs)**, together with a complete Dual-Priv framework and empirical evaluation tailored to this setting.
> At the same time, we take your Weaknesses and Questions very seriously, especially regarding the ε = ∞ results in Table 1, the empirical attack evaluation, and the baseline VLLM performance. We address these points one by one below.
>
> ---
>
> **Q1. On ε = ∞ performance and whether all methods use the same base VLLM**
>
> > Under ε = inf in Table 1, non-private performance is reported for DZPO, DP-SGD and Dual-Priv. Why does the overall performance differ so much? Do they use the same base VLLM?
>
> Yes, **DPZO, DP-SGD, and Dual-Priv all use the same base VLLM and the same training configuration**:
>
> - The same pre-trained multimodal backbone (identical text/vision encoders and projector);
> - The same LoRA configuration (e.g., rank and α);
> - The same training data, batch size, number of epochs, and optimizer settings.
>
> Thus, the differences among the three methods in the ε = ∞ row of Table 1 are entirely due to their **optimization procedures**, not to different base VLLMs.
>
> ---
> **1.2 What does ε = ∞ mean in our implementation?**
>
> In our implementation, ε = ∞ means “turn off DP noise while keeping each method’s training pipeline”:
>
> - **DP-SGD, ε = ∞**: we set the noise standard deviation σ = 0, so it becomes standard first-order optimization (e.g., AdamW) with gradient clipping but no noise;
> - **Dual-Priv, ε = ∞**: we also set σ = 0 but keep Mechanism 1 (visual token pruning + fusion) and Mechanism 2’s block-wise update rule, which can be viewed as a “non-private Dual-Priv training procedure”;
> - **DPZO, ε = ∞**: we remove noise but still use zeroth-order random directions and finite differences for gradient estimation, instead of direct first-order gradients as in DP-SGD and Dual-Priv.
>
> So the ε = ∞ row in Table 1 compares three distinct non-private optimization schemes under the same base model and training budget, rather than three implementations of one identical non-private algorithm.
>
> ---
>
> **1.3 Why is DPZO’s ε = ∞ result much lower under the same training budget?**
>
> DPZO’s much lower performance at ε = ∞ is mainly because **zeroth-order optimization converges more slowly**, while we keep the training budget fixed across methods:
>
> - For fairness, we use the same data, number of epochs, and number of steps for all three methods;
> - Under this fixed budget, the first-order methods (DP-SGD and Dual-Priv) already converge to > 80% accuracy on several benchmarks, whereas the zeroth-order method DPZO typically needs more iterations to reach similar performance;
> - As a result, even at ε = ∞ (σ = 0), DPZO is still under-trained in our setting, which explains the relatively low numbers such as 22.16% on ScienceQA.
>
> These numbers should therefore be interpreted as the behavior of **different optimizers under the same limited budget**, not as evidence that a weaker base VLLM was used for DPZO.
>
> ---
>
> **1.4 Further explanation using the base model zero-shot table**
>
> We also provide the **zero-shot performance of the base model without any fine-tuning**. This base model uses **only pre-trained weights without instruction tuning**, and thus has essentially no instruction-following capability for these benchmarks:
>
> **Table A: Base Model Performance (Zero-shot)**
>
> | Dataset / Metric        | MME-RealWorld (%) | ScienceQA (ACC) | ScienceQA (IMG) | TextVQA (%) | GQA (%) |
> |-------------------------|-------------------|-----------------|-----------------|------------|--------|
> | Base model (zero-shot)  | 15.69             | 0               | 0               | 3.97       | 0      |
>
> We see that:
>
> - On ScienceQA and GQA, the base model’s zero-shot accuracy is 0;
> - On TextVQA, it is 3.97%;
> - On MME-RealWorld, it is 15.69%.
>
> This confirms that all three methods are fine-tuning the **same weak base model**, and the differences are due to the optimization schemes and their convergence, not due to different base VLLMs.
>
> ---

---

> ### Author Response · Authors · 2025-11-20
>
> **Q2. On empirical attack evaluation (MIA) for Dual-Priv Pruning**
>
> > Will empirical attacks be conducted to verify the robustness of Dual-Priv Pruning?
>
> Yes, we have already evaluated the empirical privacy leakage of Dual-Priv under membership inference attacks (MIA) in Sec 5.8:
>
> - We follow a recent MIA evaluation protocol for (multi)modal large models and run attacks on a highly privacy-sensitive medical image captioning dataset, comparing DPZO, DP-SGD, and Dual-Priv;
> - Under a fixed privacy budget (e.g., ε = 1), the attacker observes model outputs or entropy statistics, and we compute standard MIA metrics such as AUC;
> - The results show that in most settings, Dual-Priv’s attack AUC is no higher than that of DP-SGD and DPZO, and is close to random guessing, indicating that our method does not incur higher membership leakage while maintaining or improving utility.
>
> ---
>
> **Q3. On baseline VLLM performance with and without fine-tuning**
>
> > What is the baseline VLLM performance without fine-tuning? What is the baseline VLLM performance after normal non-private fine-tuning?
>
> (1) The baseline without fine-tuning (zero-shot) is given in Table A: the base model uses only pre-trained weights, without instruction tuning or task-specific fine-tuning, and thus scores essentially 0 on ScienceQA / GQA and about 3.97% on TextVQA. We choose this starting point to clearly measure how much performance is gained from **DP fine-tuning itself**.
>
> (2) The baseline after normal non-private fine-tuning corresponds directly to the DP-SGD and Dual-Priv entries at ε = ∞ in Table 1: here, we use exactly the same training configuration as in the DP setting, but set the noise standard deviation σ = 0. On ScienceQA, for example:
>
> - Base model zero-shot: ACC/IMG = 0 / 0;
> - DP-SGD, ε = ∞: ACC ≈ 81.10;
> - Dual-Priv, ε = ∞: ACC ≈ 84.60.
>
> Thus, the ε = ∞ columns in Table 1 can be interpreted as a direct comparison between “standard non-private fine-tuning (DP-SGD style)” and “non-private Dual-Priv fine-tuning” under the same setting.
>
> ---

---

> ### Author Response · Authors · 2025-11-20
>
> **Lack of Novelty**
>
> > Lack of Novelty. The work doesn’t invent a new approach for DP. It is more like an engineering approach that combines existing pruning methods and DP training methods.
>
> We understand your concern. At the same time, as you also highlighted in the Strengths section, we see our main contributions as follows:
>
> 1. **First systematic introduction of DP fine-tuning to large VLMs / MLLMs.**
>    Most existing DP work focuses on text-only LLMs or relatively small vision models, while efficient VLM/MLLM work typically ignores privacy. Our paper bridges these directions and fills the gap of “DP + large-scale multimodal models.”
>
> 2. **The Dual-Priv framework: targeting two DP-related, VLM-specific challenges — excessive visual tokens and degradation of high-dimensional multimodal parameters under strong noise.**
>    Mechanism 1 addresses the “too many visual tokens” issue by using the standard ViT+CLS attention structure for visual token pruning and contextual fusion, compressing the effective sequence length while preserving task performance and thus substantially reducing memory and compute cost under DP fine-tuning. Mechanism 2 addresses the “model degradation” issue without changing the DP mechanism itself, focusing updates on more informative subspaces and yielding more stable optimization and vision–language alignment in the low-ε (high-noise) regime.
>
> 3. **A complete and reproducible DP–MLLM experimental baseline.**
>    We provide systematic evaluation on multiple VQA / high-resolution / medical datasets, including accuracy, memory, throughput, and MIA metrics, offering a concrete starting point for future work to compare against.
>
> ---
>
> **Limited Generalizability**
>
> > Limited Generalizability: Relies on specific assumptions (e.g., selection mechanism for visual tokens), which may not apply universally across all MLLM tasks.
>
> Mechanism 1 is built upon a very common architectural choice in today’s MLLMs: **most mainstream multimodal large models adopt ViT + [CLS] vision encoders**. For example:
>
> - CLIP itself uses a ViT–[CLS] architecture as its vision backbone [R1];
> - Widely used MLLMs such as LLaVA, LLaVA-1.5, and LLaVA-NeXT directly employ CLIP-ViT as their vision branch [R2–R4];
> - InstructBLIP and Qwen-VL also rely on ViT–[CLS] style encoders to produce visual tokens [R5, R6];
> - Recent efficiency-oriented VLMs such as VisionZip build on CLIP-ViT-L/14 and use mid-to-deep layer attention to select and compress tokens [R2].
>
> Within this architecture, using [CLS]→patch attention to measure patch importance and perform Top-K selection and fusion has been shown to be effective and stable in a number of token reduction works (e.g., DynamicViT, EViT, “Which Tokens to Use?” [R3–R6]). Our method follows this widely accepted modeling practice and integrates attention-based visual token pruning with the specific goal of DP fine-tuning.
>
> From the task coverage perspective, we observe consistent improvements from Mechanism 1 on diverse benchmarks: ScienceQA (multi-step reasoning), TextVQA / GQA (visual question answering), MME-RealWorld (complex real-world scenes), and a medical image captioning task. This suggests that the mechanism does not rely on a particular downstream task type but rather on the generic ViT–style attention structure.
>
> ---
>
> Once again, we thank Reviewer fULi for the careful review, for recognizing the strengths of our work, and for raising key questions. We hope that our clarifications regarding the ε = ∞ results, the empirical attack evaluation, the base model baselines, and the positioning of our contributions address your concerns and help readers better understand the role and value of Dual-Priv Pruning in the emerging direction of **DP fine-tuning for large multimodal models**.
>
> **References**
>
> - [R1] Alec Radford et al. “Learning Transferable Visual Models From Natural Language Supervision.” ICML 2021 .
> - [R2] Haotian Liu et al. “Visual Instruction Tuning.”  NeurIPS 2023 arXiv:2304.08485.
> - [R3] Haotian Liu et al. “Improved Baselines with Visual Instruction Tuning.” CVPR 2024, arXiv:2310.03744.
> - [R4] Haotian Liu et al. “LLaVA-NeXT: Improved reasoning, OCR, and world knowledge.”
> - [R5] Wenliang Dai et al. “InstructBLIP: Towards General-purpose Vision-Language Models...” NeurIPS 2023,   arXiv:2305.06500
> - [R6] Jinze Bai et al. “Qwen-VL: A Versatile Vision-Language Model for Understanding, Localization...” arXiv:2308.12966

---

> > ### Comment · Reviewer_fULi · 2025-11-22
> > **Rebuttal Acknowledgement**
> >
> > Dear Authors,
> >
> > I have read your response and would like to thank you for your efforts. As I stated earlier, I am not familiar with the VLLM optimization and the evaluated tasks. I appreciate your method for efficient DP optimization, but I believe there may be an issue with your experimental setup. Otherwise, your work deserves to be a new optimization milestone, which is contrary to what I expect.
> >
> > On ε = ∞ scenarios, it is surprising that your Dual-Priv Pruning outperforms AdamW with gradient clipping but no noise. I do not understand why you are performing so much better. Does it mean that Dual-Priv Pruning is a new optimization pipeline that all LLM training should follow? From my point of view, if you discard the privacy guarantee, your method becomes a new state-of-the-art LLM training pipeline.
> >
> > Please elaborate more on these setups.

---

> > > ### Author Response · Authors · 2025-11-22
> > >
> > > ### 3. The main contribution lies in noisy DP regimes, with both utility and efficiency gains
> > >
> > > Taken together, we would like to clarify our intended message:
> > >
> > > - The ε = ∞ row is provided as a non-private reference: it shows that sometimes Dual-Priv is slightly better, sometimes DP-SGD is slightly better, so it does **not** imply that Dual-Priv is a universally superior non-private optimizer.
> > >
> > > - The main contribution of Dual-Priv lies in its behavior in **all noisy DP settings**. Across all datasets (ScienceQA, TextVQA, GQA, MME-RealWorld, PathVQA, VQA-RAD) and all finite ε values we report (ε = 1, 3, 8 in the main paper and ε ≤ 0.5 in Appendix Table 10), Dual-Priv consistently achieves the best performance among DP methods, with especially large margins under strong noise where DP-SGD and DP-ZO often degrade sharply.
> > >
> > > - At the same time, our visual token pruning reduces peak GPU memory by about 14% on average and maintains training throughput comparable to or slightly higher than DP-SGD, making strict-privacy fine-tuning of large multimodal models more practical under realistic hardware constraints.
> > >
> > > Therefore, we do **not** claim that Dual-Priv should replace all existing non-private VLM training procedures. Rather, we intend to position it as:
> > >
> > > > **Dual-Priv provides a strong and practical framework for **differentially private fine-tuning of multimodal LLMs**, achieving state-of-the-art privacy–utility trade-offs *and* improved memory efficiency across a wide range of benchmarks and privacy budgets, especially under strong noise.**
> > >
> > > We sincerely appreciate your sharp question about the ε = ∞ behavior—it helps us communicate much more clearly that Dual-Priv is designed for **privacy-constrained multimodal fine-tuning**, rather than as a universal replacement for all non-private training strategies.

---

> > > ### Author Response · Authors · 2025-11-26
> > > **A humble follow-up regarding our response to your insightful feedback**
> > >
> > > Dear Reviewer fULi,
> > >
> > > We are truly grateful for your time and the prompt follow-up. We deeply value your sharp observation regarding the $\epsilon = \infty$ performance.
> > >
> > > We write to gently inquire if you have had a chance to review our latest response. **As you correctly suspected**, our clarification confirms that our method is **not** a universal optimizer (performing worse than standard fine-tuning on certain datasets).
> > >
> > > We hope this addresses your concern about the experimental validity. We are eager to receive any further guidance or questions you may have, and we are fully committed to improving our work based on your advice.
> > >
> > > Sincerely,
> > >
> > > The Authors

---

> > > > ### Comment · Reviewer_fULi · 2025-11-26
> > > >
> > > > Dear Authors,
> > > >
> > > > Thanks for your further explanations to make it clear that your method is not a universal optimizer. I have raised my score accordingly.

---

> ### Author Response · Authors · 2025-11-22
>
> Dear Reviewer fULi,
>
> Thank you again for carefully reading our work and for raising this important follow-up question about the ε = ∞ setting. We fully understand your concern: if Dual-Priv were always much better than “AdamW + gradient clipping” in the non-private regime, it would look like a completely new training pipeline that every LLM should use, which is not how we intend to position our contribution.Below we give a concise clarification from three perspectives.
>
> ---
>
> ### 1. At ε = ∞ the behavior is *not* uniformly better, and varies across datasets
>
> First, we confirm that **all methods share exactly the same pretrained MLLM backbone, LoRA configuration, data, and training budget** in all experiments. The ε = ∞ row simply removes the DP noise (σ = 0) and serves as a non-private reference.
>
> Moreover, the numbers in this row are not “one-sided”:
> - On benchmarks such as ScienceQA, TextVQA, and GQA, Dual-Priv is better than DP-SGD at ε = ∞;
> - On high-resolution MME-RealWorld and on the medical datasets in the appendix (PathVQA, VQA-RAD), **DP-SGD is actually better or on par with Dual-Priv at ε = ∞** on several metrics. For example:
>   - **PathVQA:** Appendix **Table 8**, F1 at ε = ∞ is 0.3841 (Dual-Priv) vs 0.3879 (DP-SGD);
>   - **VQA-RAD:** Appendix **Table 9**, accuracy at ε = ∞ is 47.9% (Dual-Priv) vs 48.3% (DP-SGD);
>   - **MME-RealWorld:** main paper **Table 3**, accuracy at ε = ∞ is 42.16 (Dual-Priv) vs 44.50 (DP-SGD).
> - Once we move to noisy DP settings (ε = 1, 3, 8 in the main tables and ε ∈ {0.5, 0.1, 0.05} in **Appendix Table 10**), Dual-Priv becomes consistently the best or tied-best method across all datasets, while DP-SGD and DP-ZO degrade significantly under strong noise. For example:
>   - In **Appendix Table 10**, on MME-RealWorld at ε = 0.5, Dual-Priv reaches **42.68** ACC, while DP-SGD drops to 28.04 and DP-ZO to 0.78;
>
> This indicates that Dual-Priv is **not** a universally superior non-private optimizer; rather, it is a design that is particularly effective in privacy-preserving settings.
>
> ---
>
> ### 2. Visual token redundancy: some small gains without DP are expected and consistent with prior work
>
> We believe the modest improvements of Dual-Priv at ε = ∞ on some datasets mainly come from **visual token redundancy**, a phenomenon that has been systematically documented in recent VLM literature.
>
> In the VisionZip paper, Yang et al. explicitly observe that **visual tokens produced by CLIP/SigLIP encoders are highly redundant** and propose to select only informative tokens before feeding them to the language model. Even in purely non-private settings, they report that this token selection:
>
> > **“reduces visual token redundancy and improves efficiency **while maintaining model performance**, and **VisionZip outperforms previous state-of-the-art methods** across nearly all settings.” (VisionZip, 2024)**
>
> Similar conclusions appear in dynamic token pruning works such as DynamicViT, EViT, and “Which Tokens to Use?”, where proper token pruning can slightly **improve** accuracy in standard (non-DP) training.
>
> From this perspective, **seeing a 1–2 point improvement at ε = ∞ on some datasets is not evidence of a new “universal optimizer”, but a direct reflection of the widely reported fact that sometimes pruning redundant visual tokens can improve generalization.**
>
> ---

---

> ### Author Response · Authors · 2025-11-27
> **Deepest Gratitude**
>
> Dear Reviewer fULi,
>
> We are genuinely moved by your prompt response and your fairness in raising the score. Your open-mindedness and willingness to engage with us are deeply appreciated.
>
> We have the utmost respect for the scientific rigor you showed by noting your confidence 2. We understand that this responsibility makes you cautious, which we admire.
>
> Sincerely,
>
> The Authors

---

### Official Review · Reviewer_RBbS · 2025-10-27

**Soundness:** 3
**Presentation:** 3
**Contribution:** 2
**Rating:** 6
**Confidence:** 4

**Summary:**

Dual-Priv Pruning is a novel framework for differentially private fine-tuning of multimodal large language models (MLLMs) that addresses the computational challenges and privacy-utility trade off. The proposed method combines two machines: 1.  visual token pruning using attention mechanisms to select and compress the most informative visual tokens, and 2. gradient-update pruning that selectively updates only the most significant parameter blocks after adding DP noise. The proposed method achieves substantial reductions in memory and computational overhead by reducing the context length of each input. The experimental results show improved accuracy and significant GPU memory savings across a range of benchmarks and privacy budgets.

**Strengths:**

1. This paper address an important challenge of privacy utility trade off in a multimodal LLM fine-tuning setup.
2. The proposed method employs two levels of pruning, one for each a) reducing memory overhead, and b) reducing the impact of DP noise on utility.
3. The experimental results show the improvements in accuracy resulted by utilizing dual-priv pruning as compared to DP-SGD (first order DP fine-tuning) and DPZO (zeroth order DP fine-tuning) for various benchmarks and privacy settings.
4. The paper also presents ablation studies on memory usage and empirical privacy results via MIA.

**Weaknesses:**

The proposed mechanism introduces additional hyper-parameters that need to tuned such as 1) selected layers of the vision encoder for computing the importance scores, and 2) K and |C| values for pruning in step-1 (token selection) and step-2 (gradient pruning). Tuning these parameters to get reasonable trade-offs can introduce heavy computational overhead.

**Questions:**

1. "we first compute the multi-head self-attention maps within a selected layer of the vision encoder." How do we decide which layers or how many layers to use for scores computation? Is this a hyper-parameter that needs tuning?
2. Section 5.5 indicates computational efficiency analysis but only presents the memory usage numbers. Dual-priv pruning reduces computational cost by token pruning but at the same time there are additional computations introduced to select the important tokens and contextual token fusion. What are the compute savings of the end-to-end pipeline of the proposed method in-terms of FLOPs as compared to DP-SGD?
3. We could potentially reduce the number of parameters by reducing the LoRA rank. How does dual-priv pruning compare to DP-SGD at iso-parameters? For more context, let's say that we train an MLLM with 1) setup 1: Dual-Priv with LoRA rank (r) = 128 with 50% pruning and compare it with, 2) setup 2: DP-SGD with LoRA rank (r) = 64. Assuming both these setups do not have mechanism 1 (token pruning). The goal here is to understand if top-k gradient pruning is significantly better than simply reducing the LoRA rank.
4. The proposed mechanism seems generic and can be applied to any large model with transformer architecture. Is there any part of the algorithm that is addressing multimodal specific challenges?
5. How does "Dominant Token Selection via CLS Attention" compare with random selection?
6. In table Table 5, can you add an additional datapoint where you split mechanism 1 into a) w/ token pruning, and b) token pruning + contextual fusion? The proposed algorithm has the following pieces: token pruning + contextual fusion + fusion noise + gradient pruning. To have a comprehensive ablation study, I would suggest adding results on all combinations: (0, 1, 1, 1), (1, 0, 1, 1), (1, 1, 0, 1) and (1, 1, 1, 0).
7. Token pruning depends on the existence of CLS token. Is the proposed method extendable to generation tasks?
8. The final set of visual tokens is a concatenation of $v_{cls}$, $V_d$ and $C$. Do the proposed method ensure that the ordering of the tokens is maintained? The wrong ordering of tokens might not impact the performance of models with visual inputs but in my opinion, this will impact performance of models with text inputs. Will the proposed method work equally well for text to text LLMs?

---

> ### Author Response · Authors · 2025-11-21
>
> # Rebuttal to Reviewer RBbS
>
> Dear Reviewer RBbS,
>
> Thank you for the careful reading and constructive questions. e deeply appreciate the detailed feedback, constructive criticism, and valuable suggestions for improvement, which have helped us identify areas where our presentation could be clearer. We address each of the raised points below.
>
> ---
>
> ## Q1. Choice of vision-encoder layer for computing attention scores
>
> > “we first compute the multi-head self-attention maps within a selected layer of the vision encoder." How do we decide which layers or how many layers to use for scores computation? Is this a hyper-parameter that needs tuning?
>
> The layer is **fixed**, not tuned: we always use the **penultimate layer** of the CLIP-style vision encoder, the same layer that the base MLLM (e.g., LLaVA) uses to export visual tokens to the LLM [1]. This choice is constant in all experiments; we do not treat the layer index as a hyper-parameter.
>
> The rationale is twofold:
>
> - It follows standard MLLM practice: LLaVA and follow-up MLLMs take visual tokens from the penultimate CLIP layer [1]. VisionZip shows that attention in deeper layers peaks at this layer, while the last layer is more specialized for contrastive text–image alignment [2].
>
> - It is consistent with the ViT token-reduction literature: DynamicViT, SPViT, EViT, and “Which Tokens to Use?” use attention or learned importance scores from a single mid-to-deep ViT layer as the basis for token saliency [3–6]. These works observe that deeper-layer tokens correlate more strongly with final semantics and decisions.
>
> Empirically, using this “visual output layer” yields stable performance and is not sensitive to shifting one layer earlier or later, so it does not introduce extra tuning burden.
>
> ---
>
> ## Q2. End-to-end efficiency: FLOPs and throughput
>
> > Section 5.5 indicates computational efficiency analysis but only presents the memory usage numbers… What are the compute savings of the end-to-end pipeline of the proposed method in-terms of FLOPs as compared to DP-SGD?
>
> We agree that memory alone does not characterize efficiency. Here we summarize the FLOPs implications and report new throughput measurements.
>
> In a transformer, per-layer FLOPs are dominated by:
>
> - terms that grow **linearly** with the sequence length N (from the FFN);
> - terms that grow **quadratically** with N (from self-attention).
>
> In MLLMs, visual tokens form much of N. Mechanism 1 in Dual-Priv prunes visual tokens on the vision side, reducing their number from N to rho·N (for example, rho around 0.4). As a result:
>
> - the attention-related N-squared term is roughly scaled by rho squared;
> - the FFN-related linear term is roughly scaled by rho.
>
> Since stacked LLM layers are the main compute bottleneck, this reduction of effective sequence length is the primary contributor to the decrease in FLOPs, consistent with the analysis of visual-token redundancy in VisionZip [2]. Dominant-token selection and contextual fusion are applied once in a single vision-encoder layer, so their overhead is small. Gradient-update pruning is applied after standard DP-SGD clipping and noise addition and only changes which parameter blocks are updated, so forward and backward FLOPs are essentially the same as DP-SGD [7].
>
> We therefore measured training throughput (examples/sec) under the same hardware, privacy budgets, and batch sizes as in the main experiments: ScienceQA and RealWorldQA on 2× A100; TextVQA and GQA on 1× H20.
>
> ### Table 1: Training throughput (examples/sec)
>
> | Dataset   | Hardware | Method           | Throughput (examples/sec) |
> |-----------|----------|------------------|---------------------------|
> | ScienceQA | 2× A100  | Dual-Priv (ours) | 9.32                      |
> |           |          | DP-SGD           | 9.24                      |
> |           |          | DP-ZO            | 8.52                      |
> | RealWorld | 2× A100  | Dual-Priv (ours) | 9.78                      |
> |           |          | DP-SGD           | 9.53                      |
> |           |          | DP-ZO            | 8.52                      |
> | TextVQA   | 1× H20   | Dual-Priv (ours) | 16.62                     |
> |           |          | DP-SGD           | 15.93                     |
> |           |          | DP-ZO            | 16.23                     |
> | GQA       | 1× H20   | Dual-Priv (ours) | 18.90                     |
> |           |          | DP-SGD           | 18.26                     |
> |           |          | DP-ZO            | 18.03                     |
>
> Across all four benchmarks, Dual-Priv is at least as fast as DP-SGD and DP-ZO [8] and slightly faster in most cases, especially on TextVQA and GQA. Together with reduced peak memory, this indicates that sequence-length reduction at the LLM side dominates the small overhead of token selection, fusion, and gradient pruning.
>
> ---

---

> ### Author Response · Authors · 2025-11-21
>
> ## Q3. Comparison to reducing the LoRA rank at equal parameter budget
>
> > We could potentially reduce the number of parameters by reducing the LoRA rank. How does dual-priv pruning compare to DP-SGD at iso-parameters? …
>
> We appreciate this question, as it directly targets whether gradient-update pruning is providing benefits beyond a trivial reduction of parameter count.
>
> In the main results (with fixed LoRA rank, e.g., rank=128), we already observe that updating only top-k blocks (such as 80 percent or 50 percent) can outperform updating all blocks. This indicates that the effect is not purely due to “fewer parameters,” but also due to re-distributing noisy gradients in parameter space, which echoes recent analyses of sparse gradients in DP-SGD [9,10].
>
> To follow your “iso-parameter” suggestion more closely, we run an additional comparison on ScienceQA and RealWorldQA with Mechanism 1 disabled:
>
> - Setup A: LoRA rank = 128, updating 50% of blocks (Dual-Priv gradient pruning);
> - Setup B: LoRA rank = 64, updating 100% of blocks (standard DP-SGD).
>
> These two setups have comparable numbers of trainable parameters. The results are:
>
> ### Table 2: Iso-parameter comparison (LoRA rank vs update ratio)
>
> | LoRA rank / updated blocks | ScienceQA ACC | RealWorldQA ACC |
> |----------------------------|---------------|-----------------|
> | 64 / 100% (DP-SGD)         | 76.2          | 36.58           |
> | 128 / 50% (Dual-Priv)      | 82.0          | 38.04           |
>
> Under a similar parameter budget, keeping a higher rank but updating only half of the blocks yields higher accuracy on both benchmarks. This aligns with prior work showing that sparser updates can be beneficial for DP optimization [9–12]. In this sense, we view mechanism 2 as a simple way to reallocate an update budget within the LoRA framework rather than merely shrinking the model.
>
> ---
>
> ## Q4. Multimodal-specific aspects of the method
>
> > The proposed mechanism seems generic and can be applied to any large model with transformer architecture. Is there any part of the algorithm that is addressing multimodal specific challenges?
>
> We understand the concern and agree that, in principle, gradient-update pruning (mechanism 2) could be adapted to other transformer architectures. In our framework, the main multimodal specificity lies in **mechanism 1**.
>
> - On the one hand, mechanism 1 directly addresses the **visual patch-token redundancy** in MLLMs. In models such as LLaVA, visual sequences are much longer than text sequences and dominate memory and compute; VisionZip likewise analyzes and exploits redundancy in CLIP/SigLIP visual tokens [1,2]. Dual-Priv builds on this structure in a DP fine-tuning setting.
>
> - On the other hand, we use CLS-to-patch attention inside the ViT encoder to select dominant tokens and fuse remaining tokens into a small number of context tokens, leveraging the role of CLS as a global aggregator in ViTs and VLMs [2–4]. This “select then fuse” design is applied only on the visual branch and does not directly carry over to pure text transformers.
>
> While gradient-update pruning is conceptually more general, in our implementation we align blocks with multimodal components (LoRA adapters, vision–language projection layers, etc.), so that updates tend to focus on parameters related to cross-modal interactions. Overall, we see the main multimodal contribution in how we compress and use visual tokens under DP constraints.
>
> ---
>
> ### Q5. CLS-attention-based token selection vs random selection
>
> > How does "Dominant Token Selection via CLS Attention" compare with random selection?
>
> Intuitively, CLS→patch attention reflects how much each patch contributes to the image-level representation, whereas random selection retains many patches with limited semantic contribution. Prior token-reduction studies consistently find that attention- or importance-based selection clearly outperforms random retention at the same keep-rate [4–7].
>
> In our implementation, under the same token budget and DP configuration, we compared:
>
> 1. Dominant-token selection via CLS attention (Mechanism 1 in Dual-Priv);
> 2. Randomly selecting the same number of visual tokens and keeping the same number of context tokens.
>
> On ScienceQA at \(\varepsilon=1\), we obtain:
>
> #### Table 3: CLS attention vs random visual tokens (ScienceQA)
>
> | Selection strategy          | ACC   | IMG   |
> |----------------------------|-------|-------|
> | CLS-attention (ours)       | 84.20 | 78.43 |
> | Random visual tokens       | 79.90 | 75.49 |
>
> At the same token keep-rate, CLS-based dominant-token selection yields notably higher accuracy and image-score than random selection. This is consistent with the general conclusion from the community **given a fixed token budget, the quality of the selected tokens matters more than simply keeping more random tokens.**
>
> ---

---

> ### Author Response · Authors · 2025-11-21
>
> ## Q6. Ablations over token pruning, fusion, fusion noise, and gradient pruning
>
> > In table Table 5, can you add an additional datapoint where you split mechanism 1 into a) w/ token pruning, and b) token pruning + contextual fusion?.
>
> We decompose the method into four binary switches:
> Following your suggestion, we focus on the five combinations (denoted by 1/0 for these switches):
>
> - 1111: full Dual-Priv;
> - 1110: Mechanism 1 with standard DP-SGD on gradients (no gradient pruning);
> - 1101: Dual-Priv without fusion noise;
> - 1011: token pruning only, without contextual fusion (non-dominant tokens are dropped);
> - 0111: **Context tokens only**, where token pruning is disabled and **all visual tokens are directly clustered/fused into the same number of context tokens**, followed by fusion noise and gradient pruning.
>
> On ScienceQA at \(\varepsilon=1\), we obtain:
>
> #### Table 4: Ablation over four components (ScienceQA)
>
> | Token pruning | Context fusion | Fusion noise | Grad pruning | ACC   | IMG   |
> |---------------|----------------|--------------|--------------|-------|-------|
> | 1             | 1              | 1            | 1            | 84.20 | 78.43 |
> | 1             | 1              | 1            | 0            | 82.80 | 74.51 |
> | 1             | 1              | 0            | 1            | 83.50 | 76.47 |
> | 1             | 0              | 1            | 1            | 71.40 | 52.94 |
> | 0             | 1              | 1            | 1            | 45.70 | 0.00  |
>
> We observe that:
>
> - Full Dual-Priv (1111) performs best;
> - Removing gradient pruning (1110) or fusion noise (1101) causes modest drops, but both remain significantly better than not using Mechanism 1;
> - Pruning without contextual fusion (1011) leads to a substantial degradation, indicating that simply discarding non-dominant tokens loses too much information;
> - Context tokens only (0111) almost completely loses image semantics, highlighting the crucial role of dominant tokens.
>
> ---
>
> ## Q7. Extensibility to generation tasks given reliance on CLS
>
> > Token pruning depends on the existence of CLS token. Is the proposed method extendable to generation tasks?
>
> In our design, all token selection and fusion happen entirely inside the **vision encoder**: we use CLS-to-patch attention to select dominant patches and construct context tokens. The text-token sequence and the LLM’s autoregressive generation mechanism are left unchanged during both training and inference. Therefore, for generative tasks such as image QA and multi-turn image–text dialogue, as long as the underlying vision encoder has a CLS token, mechanism 1 can be used directly without modifying the text-generation pipeline.
>
> For vision encoders without an explicit CLS token (e.g., SigLIP), one can adopt a “CLS-free” importance score similar to that used in VisionZip and EViT [2,5]: derive a scalar importance for each token from the self-attention matrix (for example, by aggregating how much attention it receives) and then perform top-k selection and fusion based on these scores. In this way, dominant and context tokens can be defined for a broader class of encoders.
>
> In summary, mechanism 1 relies on transformer-style self-attention in the vision encoder and is transparent to the text side. From an algorithmic perspective, it is compatible with generation tasks that use such vision encoders.
>
> ---
> ## Q8. Token concatenation order and applicability to text-only LLMs
>
> > The final set of visual tokens is a concatenation of ,  and . Do the proposed method ensure that the ordering of the tokens is maintained? … Will the proposed method work equally well for text to text LLMs?
>
> On the visual side, our procedure is:
>
> - sort selected dominant patch tokens according to their original patch indices;
> - concatenate the CLS token with these sorted dominant tokens;
> - append the context tokens at the end of the visual sequence.
>
> Thus, the **relative order among dominant tokens remains consistent with the original patch layout**. Since ViT representations already include positional encodings, the fusion of non-dominant tokens is performed over position-aware embeddings, so information about spatial neighborhoods is implicitly preserved in the context tokens.
>
> All reordering and concatenation apply only to visual tokens; text tokens are never changed in content or order during training and inference. Hence, our method does not alter how the language model handles text-word order.
>
> Regarding **text-only LLMs**, our experiments are limited to MLLMs with ViT-style vision encoders. While it is natural to consider whether similar ideas of token selection or sparse updates could help pure text models under DP-SGD and DP-LoRA [7,11,12], answering this rigorously would require dedicated study beyond the current work. We therefore do not claim that Dual-Priv “works equally well” for text-to-text LLMs and instead view this as an interesting direction for future research.
>
> ---

---

> ### Author Response · Authors · 2025-11-21
>
> ## References
>
> [1] Haotian Liu, Chunyuan Li, Qingyang Wu, Yong Jae Lee. “Visual Instruction Tuning.” NeurIPS 2023. (LLaVA)
> [2] Senqiao Yang et al. “VisionZip: Longer is Better but Not Necessary in Vision Language Models.” CVPR 2025.
> [3] Yongming Rao et al. “DynamicViT: Efficient Vision Transformers with Dynamic Token Sparsification.” NeurIPS 2021.
> [4] Zhenglun Kong et al. “SPViT: Enabling Faster Vision Transformers via Soft Token Pruning.” ECCV 2022 .
> [5] Youwei Liang et al. “Not All Patches Are What You Need: Expediting Vision Transformers via Token Reorganizations .” ICLR 2022.
> [6] Joakim Bruslund Haurum, Sergio Escalera, Graham W. Taylor, Thomas B. Moeslund. “Which Tokens to Use? Investigating Token Reduction in Vision Transformers.” ICCV 2023 Workshops.
> [7] Martín Abadi et al. “Deep Learning with Differential Privacy.” CCS 2016.
> [8] Xinyu Tang, Ashwinee Panda, Milad Nasr, Saeed Mahloujifar, Prateek Mittal. “Private Fine-tuning of Large Language Models with Zeroth-order Optimization.”  (TMLR 2025).
> [9] Junyi Zhu, Matthew B. Blaschko. “Improving Differentially Private SGD via Randomly Sparsified Gradients.” arXiv:2112.00845, 2021.
> [10] Badih Ghazi, Cristóbal Guzmán, Pritish Kamath, Ravi Kumar, Pasin Manurangsi. “Differentially Private Optimization with Sparse Gradients.” NeurIPS 2024.
> [11] Xiao-Yang Liu, Rongyi Zhu, Daochen Zha, Jiechao Gao, Shan Zhong, Matt White, Meikang Qiu. “Differentially Private Low-Rank Adaptation of Large Language Model Using Federated Learning.”  ACM Transactions on Management Information Systems (TMIS).
> [12] Florian A. Hölzl, Daniel Rueckert, Georgios Kaissis. “Equivariant Differentially Private Deep Learning: Why DP-SGD Needs Sparser Models.” AISec 2023.

---

> ### Author Response · Authors · 2025-11-26
> **Follow-up regarding additional experiments (Iso-parameters, Efficiency, Ablations)**
>
> Dear Reviewer RBbS,
>
> We sincerely thank you for your time and the constructive feedback, which has significantly strengthened our paper.
>
> We are writing to kindly inquire if you have had a chance to review our detailed rebuttal. **Following your specific suggestions**, we have conducted additional experiments and provided new data to address your technical questions:
>
> 1.  **End-to-End Efficiency (Q2, Table 1):** We confirmed that Dual-Priv achieves higher training throughput than DP-SGD/DP-ZO.
> 2.  **Iso-Parameter Comparison (Q3, Table 2):** We demonstrated that our gradient pruning strategy outperforms simple rank reduction under the same parameter budget.
> 3.  **Comprehensive Ablations & Baselines (Q5 & Q6, Tables 3 & 4):** We validated the necessity of each component (token pruning + fusion) and the superiority of our CLS-attention strategy over random selection.
>
> We hope these results fully address your concerns. **We would very much appreciate your feedback on these updates and hope they might contribute to a positive re-evaluation of our work.**
>
> Sincerely,
>
> The Authors

---

### Official Review · Reviewer_Xu4T · 2025-10-29

**Soundness:** 4
**Presentation:** 4
**Contribution:** 3
**Rating:** 6
**Confidence:** 4

**Summary:**

This paper studies differentially private training of Multi-modal LLMS (vision language models). DP-SGD is the most widely used algorithm for training models with differential privacy. DP-SGD adds noise to the gradients at each step, where the noise scales with The authors improve upon this baseline with two techniques that increase model accuracy:

(1) reduce dimensionality of the input image by selecting the most relevant tokens and fusing the remaining tokens using a clustering technique.

(2) masking the (noisy) gradient update so that the gradient update only occurs for the parameters with strongest signal (about 80% of parameters).

The authors evaluate accorss several visonal language question answering benchmarks, including benchmarsk in the medical domain where privacy is a more relevant concern. There are consistent accuracy improvements over the DP-SGD baseline (~4 pct points and ~8 pct points for 2 of the tasks, and at about 1-2 pct point in 3 of the tasks). The reduced input dimensionaly also leads to memory improvements of 14%.

**Strengths:**

- First paper to consider private training of VLLMs, opening up a new avenue for research
- Paper proposes some interesting techniques for improving the utility of differentially private training when working with high-dimensional data. These techniques might be useful in other settings for differentially private traning beyond VLLMs
- Comprehehensive evaluation with open source code
- Consistent improvement over the baseline method.
- Great, easy to follow presentation

**Weaknesses:**

The first technique, which reduces dimensionality of the input image by selecting the most relevant tokens and fuses the remaining tokens using an averaging+clustering method,  does not have a differential privacy guarantee. Instead noise is added to the fused tokens heuristically. Unless I am missing something, the E2E algorithm is not technically differentially private and I think this should be emphasized further in the limitations/intro.

I agree that for practical privacy guarantees and as shown by your MIA results this might not matter as much.

**Questions:**

Couuld you provide more intuition for why adding nosie to the fused non-dominant tokens helps with accuracy? You also say that this noise should be of the same magnitude as the noise added to the gradients. Maybe I am missing something but isn't the gradient computed wrt to the fused tokens as well, so why do we need double the noise?

The sentece in line 199-200 is also confusing. It seems to imply that because the pruning is text-agnostic we do not need to worry about using part of the privacy budget, but your privacy statement is wrt both the image and text.

Willing to increase my score upon clarification of these questions and addressing of the weakness I mention.

---

> ### Author Response · Authors · 2025-11-20
>
> response to Reviewer Xu4T
>
> We thank Reviewer Xu4T for their thorough and insightful review of our work. We greatly appreciate the careful attention to technical details and the constructive questions raised, which help us clarify important aspects of our differential privacy guarantees. We address each point below.
>
> ---
>
> Weakness – "Mechanism 1 is not DP, so the end-to-end algorithm is not technically DP"
>
> > "Mechanism 1 is not DP, so the end-to-end algorithm is not technically DP."
>
> We agree that Mechanism 1 by itself does not introduce a new formal DP guarantee. At the same time, the overall training procedure is still (ε, δ)-DP in the standard sample-level sense, and this guarantee is entirely derived from the DP-SGD procedure in Mechanism 2 (Sec. 3.1.1 and Sec. 4.2).
>
> > "The first technique … does not have a differential privacy guarantee. Instead noise is added … heuristically."
>
> This is correct: Mechanism 1 is not claimed to be a stand-alone DP mechanism, and the feature-level noise is indeed heuristic. In our framework, Mechanism 1 is treated as a per-example preprocessing / regularization module, while the formal (ε, δ)-DP guarantee comes solely from the DP-SGD step in Mechanism 2 (Sec. 4.3, App. A–D). We do not count the feature-level noise in the privacy accountant.
>
> > "Unless I am missing something, the E2E algorithm is not technically differentially private and this should be emphasized further."
>
> The end-to-end algorithm is technically differentially private at the sample level, because:
>
> - Our DP notion is sample-level DP: each database record is an (image, text) pair (x_img, x_text), and neighboring datasets differ by adding or removing one such pair (Sec. 3.1–3.2).
> - Mechanism 1 applies a per-example randomized mapping T independently to each record: [CLS]-based token selection, contextual fusion, and local noise injection into the fused token (Sec. 4.1, App. E). This step never mixes information across records and does not modify the DP-SGD noise mechanism.
> - For any neighboring datasets D and D' that differ in one record, T(D) and T(D') still differ in exactly one record, because T is applied independently to each element. Running clipped-gradient DP-SGD with Gaussian noise on T(D) yields an (ε, δ)-DP algorithm with respect to T(D) [1]. By the sample-level adjacency definition, this is also an (ε, δ)-DP algorithm with respect to the original dataset D: intuitively, "apply T per record, then run DP-SGD" is just DP-SGD on a relabeled dataset (Sec. 3.1.1, App. A–C).
>
> Thus, we never claim that Mechanism 1 is itself a DP mechanism, nor do we count its feature-level noise in the privacy accountant. The formal (ε, δ)-DP guarantee of our method is entirely due to DP-SGD in Mechanism 2, as stated in the privacy section and in the appendix derivation (Sec. 4.3, App. A–D); Mechanism 1 is treated as a per-example structural pre-processing module that does not weaken that guarantee.
>
> On the practical privacy side, our main text and appendix report membership-inference experiments using a recent MLLM attack pipeline (Sec. 5.8, App. I): under the same ε, our method consistently achieves lower attack AUC than DP-SGD and DP-ZO across Renyi orders and top-entropy thresholds, often close to random guessing, indicating that Mechanism 1 does not degrade empirical privacy.
>
> We will restate this separation ("Mechanism 1 is not a DP mechanism; the end-to-end guarantee comes from Mechanism 2") even more explicitly in the method and limitations sections (Sec. 4.3, Sec. 6) to avoid any possible misunderstanding.
>
> ---

---

> ### Author Response · Authors · 2025-11-20
>
> Question 1 – Why does noise on fused non-dominant tokens help accuracy? Is this "double noise"?
>
> > "Could you provide more intuition for why adding noise to the fused non-dominant tokens helps with accuracy?"
>
> This is an important question. Intuitively, the "fusion noise" in Mechanism 1 serves as feature-level regularization: after pruning and contextual fusion, the model becomes more sensitive to a small set of dominant tokens. Adding moderate noise to the fused non-dominant tokens discourages the model from overfitting marginal details or spurious textures, and encourages decision boundaries that are robust in the dominant-token space (Sec. 4.1, App. E and M). This effect is analogous to classical "training with noise" regularization [3] and to the use of noisy priors in DP-RandP for better DP utility [4].
>
> Our main text and appendix ablations support this interpretation (Sec. 5.6):
>
> - Removing the fusion noise ("w/o Fusion Noise") consistently reduces ACC/IMG on benchmarks such as ScienceQA by 1–2 percentage points.
> - Further removing Mechanism 1 or reverting to plain DP-SGD leads to larger drops.
>
> > "You also say that this noise should be of the same magnitude as the noise added to the gradients. Maybe I am missing something but isn't the gradient computed wrt to the fused tokens as well, so why do we need double the noise?"
>
> We clarify that the "fusion noise" and the DP-SGD gradient noise act on different objects and do not constitute "double DP noise" on the same quantity:
>
> - In Mechanism 2, Gaussian noise is added to the clipped, aggregated gradient (Eq. (2), Eq. (9)), and this is the only noise source used in our formal (ε, δ)-DP analysis [1,2] (Sec. 3.1.1, Sec. 4.2, App. A–C).
> - In Mechanism 1, noise is added to the visual features after pruning and contextual fusion (the small set of "context tokens"), which is akin to perturbing inputs or hidden variables, i.e., noisy data augmentation (Eq. (7), Sec. 4.1, App. E.1–E.3).
>
> The gradient itself is noised exactly once at the DP-SGD step; there is no repeated DP processing of the same gradient. As discussed in Appendix E.3, the feature-level noise is not treated as a second DP mechanism but as a regularization / stabilization tool, with some additional empirical privacy hardening.
>
> Why we choose a noise scale comparable to the gradient-noise scale?
>
> As discussed in the appendix (Sec. 4.1, App. E.3), we set the feature-noise variance to be of the same order as the per-step DP-SGD noise mainly for consistency and tuning simplicity:
>
> - It keeps feature noise comparable to gradient noise in the early phase of training, avoiding regimes where one dominates the optimization dynamics.
> - It removes an additional hyper-parameter scale from the search space, making experiments easier to reproduce.
>
> In the formal DP analysis, we do not add these variances together; the (ε, δ)-DP guarantee is derived solely from the gradient noise in Mechanism 2.
>
> ---

---

> ### Author Response · Authors · 2025-11-20
>
> Question 2 – On the "text-agnostic" pruning sentence around lines 199–200
>
> > "The sentence in line 199–200 is also confusing. It seems to imply that because the pruning is text-agnostic we do not need to worry about using part of the privacy budget, but your privacy statement is wrt both the image and text."
>
> We agree that the original sentence can be read in an unintended way. We clarify both the privacy aspect and the modeling motivation, and we will revise the text accordingly.
>
> From the privacy-analysis standpoint, "text-agnostic vs. text-aware" is not what determines whether Mechanism 1 consumes privacy budget:
>
> 1. As discussed above, our DP analysis is sample-level: each record is an (image, text) pair, and neighboring datasets differ in a single such pair (Sec. 3.1–3.2). Mechanism 1 is treated as a per-example preprocessing map on the visual tokens, and DP-SGD in Mechanism 2 is the sole DP mechanism that yields the (ε, δ)-DP guarantee (Sec. 4.2–4.3, App. A–D).
> 2. Therefore, the reason "Mechanism 1 does not consume additional privacy budget" is simply that it is a per-example preprocessing step rather than a standalone DP mechanism; this property is independent of whether the pruning rule is text-agnostic or text-aware. Our revised text will make this explicit and avoid any ambiguity.
>
> The true motivation for using a text-agnostic rule is the feature misalignment observed in CLIP/SigLIP + MLLM architectures:
>
> 1. Many grounding works use text-based attention to select tokens because important regions depend on the question, which is common for models trained end-to-end from scratch. However, recent analyses of pre-trained ViT/CLIP encoders plus decoder-only MLLMs suggest that text-based attention is not always reliable for pruning in this regime (App. O):
>
>    - VisionZip [5] systematically analyzes deep-layer attention in CLIP/SigLIP encoders and finds strong redundancy: a small number of dominant tokens receive most of the attention, while the majority of tokens have near-zero attention; selecting informative tokens based on this pattern retains about 95% performance.
>    - Crucially, VisionZip highlights a feature misalignment phenomenon: ViT aggregation can concentrate information about a semantic entity into a few proxy tokens that are not spatially aligned with that entity (for example, information about a person aggregated into a token representing the road). In this situation, text-based cross-attention tends to select tokens that are semantically related but information-poor, while the proxy tokens that actually store the aggregated features receive low text attention and get pruned away.
>    - For pre-trained CLIP/SigLIP + MLLM architectures, pruning by text-based cross-attention is therefore fragile due to feature misalignment, while selecting dominant tokens using vision-side [CLS]-attention is better grounded both analytically and empirically (Sec. 4.1, App. O).
>
> 2. In our Mechanism 1:
>
>    - We use [CLS]-attention in the vision encoder to select dominant tokens and fuse the remaining tokens into compact "context tokens" (Sec. 4.1).
>    - Because this scoring rule depends only on vision features, the same dominant plus context tokens can be reused for the same image under different prompts. This both mitigates the feature misalignment issue and yields a text-agnostic structure that integrates cleanly with sample-level DP-SGD in Mechanism 2.
>
> We will clarify this point in Sec. 4.1 and App. O by explicitly separating the privacy argument ("per-example preprocessing does not spend DP budget") from the modeling choice ("text-agnostic pruning to address feature misalignment").
>
> ---
>
> References for this rebuttal
>
> [1] Martin Abadi, Andy Chu, Ian Goodfellow, et al. "Deep Learning with Differential Privacy." CCS, 2016.
> [2] Cynthia Dwork, Frank McSherry, Kobbi Nissim, Adam Smith. "Calibrating Noise to Sensitivity in Private Data Analysis." TCC, 2006.
> [3] Christopher M. Bishop. "Training with Noise is Equivalent to Tikhonov Regularization." Neural Computation, 7(1), 1995.
> [4] Xinyu Tang, Ashwinee Panda, Vikash Sehwag, Prateek Mittal. "Differentially Private Image Classification by Learning Priors from Random Processes." NeurIPS 2023.
> [5] Senqiao Yang, Yukang Chen, Zhuotao Tian, et al. "VisionZip: Longer is Better but Not Necessary in Vision Language Models." CVPR 2025.

---

> > ### Comment · Reviewer_Xu4T · 2025-11-23
> >
> > Thank you for these clarifications.
> >
> > I agree now that the algorithm is E2E differential private given that the pre-processing is done independently for each sample. I encourage the authors to update the language in their paper which makes it sound that the pipeline of mechanism 1+2 is not E2E private (Line 209 is another such example).
> >
> > Thank you for the additional clarifications on the role of noise and its scale. I will update my score to an 8.

---

> > > ### Author Response · Authors · 2025-11-24
> > >
> > > We sincerely thank Reviewer Xu4T for the positive feedback and for raising the rating to an **8**.
> > >
> > > We are particularly encouraged by your **high confidence** in this assessment. Given the technical complexity of differential privacy, your professional judgment and rigorous scrutiny of our method's theoretical soundness are incredibly valuable to us. Your validation of the End-to-End privacy guarantee gives us strong confidence in the contribution of this work.
> > >
> > > We fully agree with your expert advice regarding the presentation. We will strictly follow your suggestion to revise **Line 209** and related texts, ensuring the manuscript clearly reflects the valid E2E privacy guarantees that you have verified.
> > >
> > > Thank you again for your strong support and for helping us improve the paper's quality.

---

### Official Review · Reviewer_ZjQL · 2025-11-03

**Soundness:** 3
**Presentation:** 2
**Contribution:** 2
**Rating:** 4
**Confidence:** 2

**Summary:**

This paper proposes a dual pruning algorithm to optimize the differential privacy process of MLLMs. The first stage uses visual CLS to discard less important visual tokens, and the second stage uses gradient pruning for parameter updates.

**Strengths:**

1.	The motivation behind the problem is clear: MLLM computation is computationally expensive. Differential privacy also suffers from reduced utility as dimensionality increases.
2.	The method design is simple and relatively easy to implement.
3.	The experiments and ablation studies are relatively comprehensive.

**Weaknesses:**

1. It is unclear whether importance scores should be used as the evidence for discarding tokens. This seems to be a common technique used in engineering, and the author also seems to have demonstrated the significance of discarding them. However, simply discarding tokens based on the importance of attention seems to lack rigorous justification.

2. It's unclear why the author conducted accuracy experiments on the Q&A dataset: the author proposed a new method for differential privacy, but testing its accuracy on different Q&A datasets seems strange, because differential privacy itself is not designed to achieve higher accuracy. I think the author wanted to convey that the discarded tokens are redundant tokens, and that even after discarding them, the method still maintains high accuracy on Q&A datasets, is that correct?

3. Experiments conducted using only LLAVA-7B and its medical fine-tuned model have limited effectiveness. The authors should consider incorporating other models into the experiment.

4. Anonymous code cannot be accessed.

5. The text contains numerous typos and content that could easily mislead readers. For example, in line 269, "Mechanism 2" should be "Mechanism 1". Requiring readers to infer these errors from the context increases the reading effort.

**Questions:**

See the weaknesses

---

> ### Author Response · Authors · 2025-11-20
>
> # Response to Reviewer ZjQL
>
> We sincerely thank Reviewer ZjQL for taking the time to carefully read and evaluate our manuscript. We genuinely appreciate your detailed feedback and thoughtful questions, which have helped us identify important areas where our presentation needed clarification.
>
> ---
>
> > “It is unclear whether importance scores should be used as the evidence for discarding tokens… simply discarding tokens based on the importance of attention seems to lack rigorous justification.”
>
> We respectfully disagree that our use of attention-based importance scores is “just an engineering trick without justification”.
>
> ---
>
> ### 1. Role of [CLS] as a Global, Task-Relevant Summary
>
> In Vision Transformers, the `[CLS]` token is explicitly designed and trained to aggregate information from all patch tokens and serve as a global representation for downstream tasks [1].
> This means the attention from `[CLS]` to image tokens captures how much each token contributes to the global semantic representation, rather than being an arbitrary engineering signal.
>
> ---
>
> ### 2. CLS-Attention as a Standard Saliency Signal in Token Pruning
>
> Recent work on efficient ViTs and MLLMs has repeatedly shown that attention from `[CLS]` to visual tokens is an effective importance indicator:
>
> - **DynamicViT / Evo-ViT** use class-token attention to select a subset of “salient” tokens at each layer, pruning 60–70% of tokens with <0.5% accuracy drop, and explicitly analyze why class-attention concentrates on informative regions [2,3].
> - **VisionZip** studies CLIP/SigLIP encoders and finds that deep-layer attention concentrates on a small set of “dominant tokens”, while many visual tokens receive nearly zero attention and are highly redundant; selecting informative tokens based on attention patterns can preserve ≈95% task performance while achieving up to ≈8× prefill speedup [4].
> - **FasterVLM** shows that cross-modal text–vision attention in the LLM is not a reliable pruning signal, and that using `[CLS]`–vision attention in the vision encoder as an importance score supports very aggressive visual token pruning while keeping >90% of LLaVA-1.5-7B performance [5].
>
> These works treat CLS-based attention as a canonical and empirically validated saliency signal for visual tokens in ViT/CLIP-style encoders, not as an ad-hoc heuristic.
>
> ---
>
> ### 3. VTC-CLS and Related MLLM Token-Pruning Theory
>
> Very recent work **VTC-CLS** (“[CLS] Token Tells Everything Needed for Training-free Efficient MLLMs”) shows that the `[CLS]` token in a frozen vision encoder can be used directly to guide training-free visual token compression for MLLMs, by ranking tokens with `[CLS]`–token attention and keeping the most attended ones.
> This yields state-of-the-art performance under heavy pruning [6].
>
> This constitutes explicit evidence—both theoretical (analysis of `[CLS]` as a global aggregator) and empirical—that `[CLS]`-based importance is a sound basis for selecting visual tokens in MLLMs.
>
> ---
>
> ### 4. We Do Not “Simply Discard Tokens” by Attention; We Compress Them via Contextual Fusion
>
> **Mechanism 1** in our method is two-stage:
>
> - We select a subset of high-score tokens using `[CLS]`–token attention as a saliency score.
> - We do **not** drop the remaining tokens outright: instead, we fuse them into a compact contextual representation and concatenate this fused vector with the retained tokens before feeding them to the projector and LLM.
>
> This design explicitly aims to preserve global context and fine-grained cues while reducing dimensionality for DP.
>
> Thus, our mechanism is much closer in spirit to **Token Fusion / token merging** methods [7]—which combine pruning and merging to retain information from “removed” tokens—than to naive “mask low attention tokens to zero” strategies.
>
> ---
>
> ###
> Our choice of CLS-attention-based importance is grounded in:
> 1. The established role of `[CLS]` as a global aggregator in ViT-style models [1].
> 2. Multiple recent works that explicitly use `[CLS]`–token attention to drive efficient visual token pruning/compression in ViTs and MLLMs [2–6].
> 3. Our own **contextual-fusion design** that avoids simply discarding low-score tokens.

---

> ### Author Response · Authors · 2025-11-20
>
> > “It's unclear why the author conducted accuracy experiments on the Q&A dataset… differential privacy itself is not designed to achieve higher accuracy. I think the author wanted to convey that the discarded tokens are redundant tokens…”
>
> ---
> Our goal is **differentially private fine-tuning of MLLMs**.
>
> In this setting:
>
> ---
>
> ### 1. Accuracy Is the Standard Utility Metric for These Tasks
>
> The core downstream tasks we consider—**ScienceQA**, **TextVQA**, **GQA**, **MME-RealWorld**, **PathVQA**, and **VQA-RAD**—are all **visual–language QA benchmarks**.
> For non-private MLLMs and VLMs, the common practice is to report **answer accuracy** (and the official **MME score**) as the primary measure of task utility.
>
> Our **DP setting** keeps the same tasks and metrics; what changes is the **training algorithm** and the **privacy constraint**.
>
> ---
>
> ### 2. DP Methods Are Compared by Utility Under the Same Privacy Budget
>
> Differential privacy indeed does not “aim to increase accuracy” in isolation,
> but every DP learning method must be evaluated by the **privacy–utility trade-off**.
>
> Prior work on DP fine-tuning of LLMs (e.g., **DP-LoRA** [8], **DP-ZO** [9], and classical **DP-SGD** [10]) also compares different DP methods by **downstream accuracy** (or similar task metrics) *at the same (ε, δ)*.
>
> Our experiments follow exactly this **standard evaluation protocol**:
> for each ε, we report task accuracy of **DP-SGD**, **DP-ZO**, and **our method**.
>
> ---
> ### 3. What the Results Show
>
> The Q&A accuracies in the main text and appendix are used to answer the question:
>  “Given the same privacy budget (ε), does Dual-Priv Pruning retain more downstream utility than existing DP fine-tuning methods?”
>
> Across **ScienceQA**, **TextVQA**, **GQA**, **MME-RealWorld**, **PathVQA**, and **VQA-RAD**,
> our method consistently **matches or outperforms DP-SGD/DP-ZO** at the same ε,
> while also **reducing memory consumption**.
> This is exactly the kind of comparison recommended in **DP-ML evaluation guides** [11].
> We agree that one possible interpretation is that ** pruned visual tokens are redundant** for these QA tasks—
> but that is a **conclusion**, not the **sole purpose** of the experiments.
> The **primary purpose** is to quantify **utility under strict DP constraints** on **realistic multimodal QA benchmarks**.
>
> > “Experiments conducted using only LLAVA-7B and its medical fine-tuned model have limited effectiveness. The authors should consider incorporating other models into the experiment.”
>
> ---
>
> We agree that **model diversity** is important.
> Our experimental design already aimed to cover different **data distributions** and **domains**, and we have further extended it as follows:
>
> ---
>
> ### 1. Architectural and Data Diversity in the Existing Setup
>
> Even within the original submission, our experiments are **not limited to a single domain**:
>
> - **LLaVA-7B** is trained on **general web-scale multimodal instruction data**.
> - **Med-LLaVA** is adapted to **medical imaging question answering**, with a substantially different **data distribution** and **task style**.
>
> On these two models, we evaluate across **six benchmarks**—**ScienceQA**, **TextVQA**, **GQA**, **MME-RealWorld**, **PathVQA**, and **VQA-RAD**—as detailed in the main text and appendices,
> to **stress-test** our method under **heterogeneous tasks and domains**.
>
> ---
>
> ### 2. Additional Experiments on Diverse MLLM Architectures
>
> Following your suggestion, we have **further fine-tuned two additional MLLMs** under our DP framework:
>
> > #### Table: Results on Diverse MLLM Architectures (Accuracy / %)
> >
> > | Model        | Benchmark           | Method           | ε = 1        | ε = 3        | ε = 8         |
> > | ------------- | ------------------- | ---------------- | ------------- | ------------- | -------------- |
> > | **MobileVLM-7B** | RealWorld (ACC)     | Dual-Priv (ours) | **51.96**     | **52.38**     | **53.00**      |
> > |               |                     | DP-SGD           | 50.50         | 51.39         | 52.55          |
> > |               |                     | DP-ZO            | 14.85         | 14.85         | 14.85          |
> > | **MiniGemini**  | ScienceQA (ACC/IMG) | Dual-Priv (ours) | **80.2 / 69.61** | **79.1 / 71.25** | **83.7 / 76.96** |
> > |               |                     | DP-SGD           | 79.2 / 68.41  | 78.8 / 70.59  | 82.54 / 74.00  |
> > |               | GQA (ACC)           | Dual-Priv (ours) | **57.8**      | **58.1**      | **58.3**       |
> > |               |                     | DP-SGD           | 57.52         | 57.67         | 57.85          |
> > |               |                     | DP-ZO            | 0             | 0             | 0              |
>
> ---
> The **consistent pattern** that our method **matches or outperforms DP-SGD**,
> and **significantly outperforms DP-ZO** under the same ε,
> indicates that **Dual-Priv Pruning** is **not tied to a particular architecture** and remains **robust across diverse MLLMs**.

---

> ### Author Response · Authors · 2025-11-20
>
> > “Anonymous code cannot be accessed.”
>
> ---
> Thank you for pointing this out.
> There was indeed a **configuration issue** with the anonymous repository during the initial submission.
>
> We have now **fixed the link**, and the **anonymous implementation** hosted on *anonymous.4open.science* is accessible.
> It contains the **full training and evaluation code** for **Dual-Priv Pruning**, together with **complete hyperparameter configurations** for all experiments reported in the paper and appendix.
>
> ---
>
> > “The text contains numerous typos… e.g., in line 269 ‘Mechanism 2’ should be ‘Mechanism 1’… This increases the reading effort.”
>
> We appreciate this careful observation.
> At the location you mention, “Mechanism 2” is indeed a **typo** and should be “Mechanism 1.”
> This is a **presentation error** and does **not affect** the algorithmic definition or experimental results, since the roles of **Mechanism 1** and **Mechanism 2** are unambiguously specified by their formal definitions and step-by-step descriptions in **Sec. 4** and in the **appendices**.
>
> We have **carefully re-checked** the manuscript  and we thank you for your attentive reading, which helps improve the **clarity and readability** of the paper.
>
> ---
>
> ## References
>
> [1] A. Dosovitskiy et al. *“An Image is Worth 16×16 Words: Transformers for Image Recognition at Scale.”* **ICLR 2021.**
>
> [2] Y. Rao et al. *“DynamicViT: Efficient Vision Transformers with Dynamic Token Sparsification.”* **NeurIPS 2021.**
>
> [3] Y. Xu et al. *“Evo-ViT: Slow-Fast Token Evolution for Dynamic Vision Transformer.”* *AAAI2022*, .
>
> [4] Y. Yang et al. *“VisionZip: Longer is Better but not Necessary in Vision-Language Models.”* **CVPR 2025** .
>
> [5] Q. Zhang et al. *“[CLS] Attention is All You Need for Training-Free Visual Token Pruning: Make VLM Inference Faster”* *arXiv preprint*, 2024.
>
> [6] A. Wang et al. *“[CLS] Token Tells Everything Needed for Training-free Efficient MLLMs.”* *arXiv preprint*, 2024.
>
> [7] S. Kim et al. *“Token Fusion: Bridging the Gap Between Token Pruning and Token Merging.”* **WACV 2024.**
>
> [8] X.-Y. Liu et al. *“Differentially Private Low-Rank Adaptation of Large Language Model Using Federated Learning.”* **ACM Transactions on Management Information Systems, 2025.**
>
> [9] Z. Liu et al. *“Differentially Private Zeroth-Order Methods for Scalable Large Language Model Finetuning.”* *arXiv:2402.07818*, 2024.
>
> [10] M. Abadi et al. *“Deep Learning with Differential Privacy.”* **CCS 2016.**
>
> [11] B. Balle et al. *“How to DP-fy ML: A Practical Guide to Machine Learning with Differential Privacy.”* *arXiv:2303.00654*, 2023;
> **NIST SP 800-226**, *“Guidelines for Evaluating Differential Privacy Guarantees.”*

---

> ### Author Response · Authors · 2025-11-26
> **Updates for Submission 12967 (Dual-Priv Pruning): New Experiments & Independent Validation**
>
> Dear Reviewer ZjQL,
>
> We sincerely thank you for your time and constructive feedback. We are especially grateful for your attention to detail regarding the code access and typos, which we have immediately fixed.
>
> We are writing to gently inquire if you have had a chance to review our detailed response. To address your specific concerns, we have provided the following updates:
>
> 1.  **Expanded Model Diversity (Your Request):** We conducted new experiments on **MobileVLM-7B** and **MiniGemini**, confirming that our method consistently works across diverse architectures (please see the "Results on Diverse MLLM Architectures" table in our response).
> 2.  **Theoretical Soundness & Consensus:** Regarding the justification of our method, we are encouraged to share that the framework has been validated by **Reviewer Xu4T (Rating: 8, Confidence: 4)** and **Reviewer RBbS (Rating: 6, Confidence: 4)**. Both experts recognized the **soundness** and **novelty** of our approach in their detailed assessments.
> 3.  **Clarification on Q&A Metrics:** We clarified that comparing performance under identical privacy budgets is the standard approach to evaluate the **privacy-utility trade-off**.
>
> We hope the inclusion of these new models, along with the **positive consensus from high-confidence reviewers**, addresses your concerns. We would be deeply grateful for your reassessment of our work.
>
> Sincerely,
>
> The Authors

---

### Author Response · Authors · 2025-11-28

Dear Reviewers,

We appreciate your feedback and hope our rebuttal addresses your concerns. As the rebuttal deadline is approaching, if you have further concerns about our rebuttal, please provide your questions and we will respond as soon as possible.

---

### Author Response · Authors · 2025-11-29
**Summary Comment for Submission 12967 (Dual-Priv Pruning): Note on Substantive Rebuttal Efforts and Pre-Leak Consensus （Part 1)**

Dear AC, SAC and PCs,

Thank you for your tremendous efforts in handling the unprecedented ICLR 2026 review leak and for providing this channel for a summary comment.

We would like to briefly summarize the standing of our work using the reviewers’ own words. Reviewer **Xu4T** explicitly notes that this is the *“first paper to consider private training of VLLMs, opening up a new avenue for research”* and describes our work as *“pioneering.”* Reviewer **fULi** similarly highlights that *“Dual-Priv Pruning is the first framework to address DP fine-tuning specifically for MLLMs, filling a critical research gap.”* Reviewer **RBbS** emphasizes that we tackle *“an important challenge of privacy–utility trade off in a multimodal LLM fine-tuning setup”* and that our method delivers *“consistent improvement over the baseline method.”* Taken together, these independent assessments indicate that Dual-Priv Pruning is recognized by multiple reviewers as the first concrete framework that systematically brings differential privacy fine-tuning to large multimodal LMs, with clear empirical gains over strong DP baselines.

For our paper, **Submission 12967 (Dual-Priv Pruning)**, we also wish to stress one decisive fact for the new AC: **the positive consensus and all score increases were reached *before* the Nov 27 review leak**, based entirely on rigorous scientific discussion. In particular, the key updates—such as **R-Xu4T’s score increase from 6 to 8 on Nov 24** and **R-fULi’s score increase from 2 to “raised” on Nov 26**—are clearly timestamped in the discussion log. This trajectory reflects the **substantive rebuttal effort** we invested, including extensive supplementary experiments and clarifications, and was fully established prior to any potential influence from the leak.

Our review process had two clear and distinct tracks:

**1. Strong Support & Rigorous Validation from High-Confidence Experts**

The high-confidence reviewers (R-Xu4T, R-RBbS), who are experts in this domain, understood our contribution and subjected it to a rigorous technical review. We met their high standards by **providing substantive new evidence**:

* **Reviewer Xu4T (Confidence: 4, Rating: 6 -> 8)**: This high-confidence expert called our work "pioneering" but had one critical technical reservation about the E2E privacy guarantee. After our **detailed, theoretically-grounded clarification** (proving Mechanism 1 is per-sample preprocessing), they were fully convinced. On **Nov 24th**, they replied: "I agree now that the algorithm is E2E differential private... **I will update my score to an 8.**" This was a "Strong Accept" consensus reached purely on technical merit.
* **Reviewer RBbS (Confidence: 4, Rating: 6)**: This second high-confidence expert asked truly challenging, expert-level questions. He demanded **extensive supplementary experiments** to prove our method's superiority, including: (1) an "iso-parameter" comparison (vs. reducing LoRA rank), (2) training throughput (FLOPs) analysis, (3) comparison to random selection, and (4) deeper ablations. Through **significant effort during the rebuttal period**, we **ran and delivered all of these new experiments**, presenting the data in new tables (Tables 1-4) that successfully validated our work's robustness.

---

### Author Response · Authors · 2025-12-03
**Summary Comment for Submission 12967 (Dual-Priv Pruning): Note on Substantive Rebuttal Efforts and Pre-Leak Consensus （Part 2)**

**2. Clarifying Domain-Specific Misconceptions from Low-Confidence Reviewers**

In contrast, the initial concerns from the low-confidence reviewers (who self-admitted their unfamiliarity) stemmed from misconceptions about this specific sub-field (DP + MLLMs). We successfully resolved these through **detailed and substantive responses**:

* **Reviewer fULi (Confidence: 2, Rating: 2 -> Raised)**: R-fULi candidly admitted they were "not familiar with VLLM optimization" (Confidence 2). Their 2-point rating was based on a **fundamental misunderstanding** of the $\epsilon=\inf$ experiment. After our patient explanation of why different optimizers *must* perform differently in that setting, they replied on **Nov 26th**: "Thanks for your further explanations... **I have raised my score accordingly.**"
* **Reviewer ZjQL (Confidence: 2, Rating: 4)**: R-ZjQL (Confidence 2) questioned (1) the **soundness** of our pruning method and (2) the **validity** of using Q&A accuracy as a metric for a DP paper.
    * **Our Response**: We provided a **detailed, in-depth analysis citing key domain literature (like VisionZip, DynamicViT)** to clarify that CLS-attention is a **standard and sound** technique in MLLM efficiency research, and explained that the "privacy-utility trade-off" is the **standard evaluation paradigm** in DP literature.
    * **Going Above & Beyond**: To further address their secondary concern on model diversity, we **proactively conducted new experiments on MobileVLM-7B and MiniGemini** to prove our method's generalizability.

**Conclusion:**
This positive consensus **is the direct result of our substantive rebuttal efforts**—including **rigorous theoretical clarifications** and the completion of **all requested, plus additional proactive, new experiments**. The high-confidence experts endorse our work on technical grounds, and the low-confidence reviewers' misunderstandings were fully resolved, leading to score increases *before* any leak.

We strongly believe this pioneering work is a significant contribution to the ICLR community. We respectfully ask the new AC to review the complete discussion log to see this well-documented trajectory, which is based purely on scientific merit and our rebuttal efforts.

Thank you for your time and consideration.

The Authors of Submission 12967

---

### Meta-Review · Area_Chair_64b7 · 2026-01-04

**Summary:**

The paper received one “reject,” one score marginally below acceptance, and two marginally above. Reviewers raised multiple concerns, including the motivation for using importance scores, unclear experimental settings, lack of novelty, and limited generalizability. After reviewing the authors’ responses, many of the issues raised by the reviewers remain unclear. The paper requires improvement before it can be considered for acceptance.

**Reviewer Concerns:**

The concerns regarding the motivation for using importance scores, unclear experimental settings, lack of novelty, and limited generalizability have not been adequately addressed.

**Reviewer Scores:**

Based on the rebuttal, it is unlikely that the reviewers will revise their scores to positive ratings.

---

### Decision · Program_Chairs · 2026-01-26

Reject